# Deep Random Features for Scalable Interpolation of Spatiotemporal Data

**Weibin Chen**[1]**, Azhir Mahmood**[1,2]**, Michel Tsamados**[1] **& So Takao**[3]
[1]University College London, UK
[2]PhysicsX, UK
[3]California Institute of Technology, USA

## Abstract

The rapid growth of earth observation systems calls for a scalable approach to interpolate remote-sensing observations. These methods in principle, should acquire more information about the observed field as data grows. Gaussian processes (GPs) are candidate model choices for interpolation. However, due to their poor scalability, they usually rely on inducing points for inference, which restricts their expressivity. Moreover, commonly imposed assumptions such as stationarity prevents them from capturing complex patterns in the data. While deep GPs can overcome this issue, training and making inference with them are difficult, again requiring crude approximations via inducing points. In this work, we instead approach the problem through Bayesian deep learning, where spatiotemporal fields are represented by deep neural networks, whose layers share the inductive bias of stationary GPs on the plane/sphere via random feature expansions. This allows one to (1) capture high frequency patterns in the data, and (2) use mini-batched gradient descent for large scale training. We experiment on various remote sensing data at local/global scales, showing that our approach produce competitive or superior results to existing methods, with well-calibrated uncertainties.

## 1 Introduction

The advent of earth observation systems have made it possible to monitor virtually all of earth's atmosphere and the ocean at unprecedented scales. This development has been pivotal to the understanding of anthropogenic impact on the environment, including global warming and rise in sea level. Hence, it is crucial that we are able to process the voluminous data effectively and extract maximal information from it to make better informed decisions in our path to achieving sustainable development goals.

However, observations from satellite products are inherently sparse in space-time, requiring methods to effectively fill in the gap at unobserved locations (Le Traon et al., 1998). This typically relies on data assimilation techniques such as the ensemble Kalman filter (Evensen, 2003), which requires one to have access to a physical model that describes the evolution of the field. While this can produce detailed and accurate reconstructions of the field, the physical models are typically expensive to run at high resolutions, often requiring access to high performance compute clusters. This can be challenging when one does not have the expertise nor the resources to gain access and/or run the models. On the other hand, statistical methods such as Gaussian process regression (GPR, Williams & Rasmussen (2006)) can be deployed. However, GPR scales poorly to large data sets, necessitating approximate inference schemes such as sparse Gaussian processes (Titsias, 2009), which may result in crude approximations if the underlying process does not have sufficiently large lengthscale or smoothness (Burt et al., 2019). Moreover, kernels used for GPR are often too simplistic, which can prevent learning of detailed fluctuations in the underlying non-stationary and multi-scale field. Deep Gaussian processes (DGPs) (Damianou & Lawrence, 2013) have emerged as an attractive solution to the latter problem. However, they still suffer from the difficulty of computing the posterior, again requiring variational inference to learn only a crude approximation to the true posterior.

In recent years, Bayesian deep learning (BDL) have emerged as an alternative paradigm for statistical modelling, which combines the flexibility and scalability of deep learning methods with

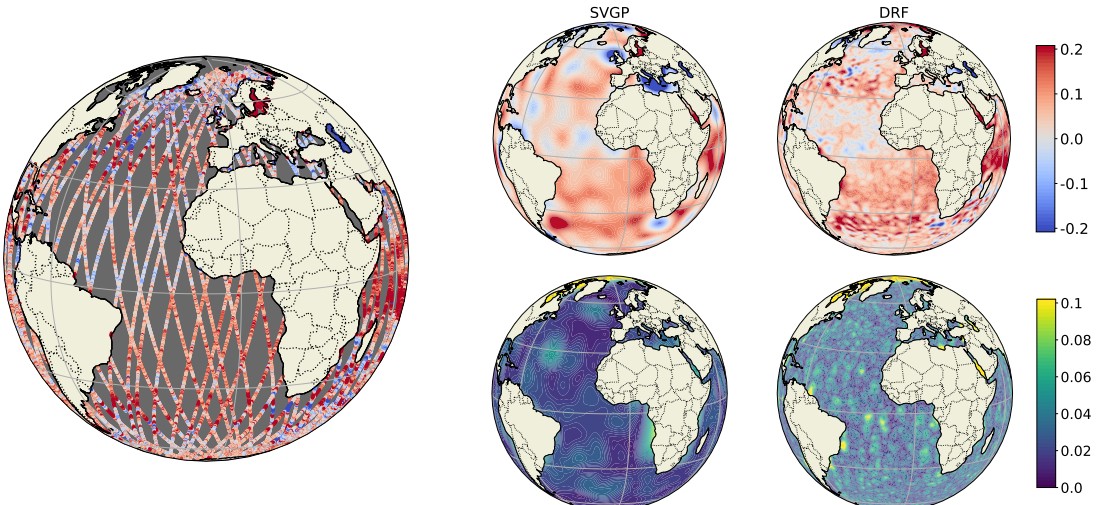

Figure 1: We propose deep random features (DRF) for accurate and flexible interpolation of satellite measurements of the earth's surface (Left). Compared to sparse variational GPs (SVGP, Centre), an ensemble of DRFs is able to achieve more detailed reconstructions of the field with sensible uncertainty estimates (Right).

Bayesian modelling principles (Papamarkou et al., 2024). In our current setting, we can approach spatiotemporal interpolation using BDL, by representing the ground truth underlying field $f^\dagger$ by a Bayesian neural network $f_\theta : \mathcal{X} \to \mathcal{Y}$, (here, $\mathcal{X}$ denotes the spatiotemporal input space and $\mathcal{Y}$ the output signals) and training on input-output pairs $\mathcal{D} = \{(x_n, y_n)\}_{n=1}^N$ for $x_n \in \mathcal{X}$ and $y_n \in \mathcal{Y}$, corresponding to earth observations. However, naïve design choices for $f_\theta$ can lead to poor reconstructions of $f^\dagger$; for example, a vanilla deep ReLU network is bound to perform poorly as it fails to learn high frequency features Tancik et al. (2020). On the other hand, deep neural networks with trigonometric activations (Sitzmann et al., 2020; Lu & Shafto, 2022) have emerged as an effective model for representing high-frequency spatiotemporal signals. However, they are not designed for interpolation of sparse data in mind and are therefore prone to overfitting.

Taken altogether, we propose to design $f_\theta$ inspired by DGPs, such that it retains the learning capacity of DNNs, while having the interpretability and inductive biases of GPs. Our main contributions are as follows: We propose the use of kernel-derived random features (Rahimi & Recht, 2007) as building blocks for BNNs to model spatiotemporal fields. We demonstrate through extensive experiments that they are capable of capturing fine-scale information in data, while being able to quantify uncertainty accurately by considering deep ensembles. Furthermore, motivated by recent developments in geometric probabilistic modelling (Borovitskiy et al., 2020), we also consider analogous random features on the sphere, leading to a novel DNN architecture with Gegenbauer polynomial activation functions that can model *global* weather fields that are adapted to the sphere. Our models are easily implementable in modern deep learning frameworks such as `PyTorch` and scale up to large datasets exceeding millions of data points through mini-batched gradient-based optimisation, pushing the boundary of what is currently possible with statistical interpolation.

## 1.1 RELATED WORKS

In Cutajar et al. (2017), trigonometric feature expansion of DGPs similar to ours have been considered, with the intent of proposing a tractable variational inference (VI) scheme for DGPs. They show superior performance to mean-field VI (Damianou & Lawrence, 2013), however, have been largely overlooked due in part to the adoption of doubly stochastic VI (Salimbeni & Deisenroth, 2017), as the de facto standard method for DGP inference. Jiang et al. (2024) similarly leverages random Fourier features to approximate kernel machines, emphasising composite kernels for incorporating prior knowledge into neural networks. DNNs with trigonometric activations have resurfaced as an object of interest more recently, with the emergence of neural radiance fields (Mildenhall et al., 2021) and subsequent work on implicit neural representations Tancik et al. (2020); Sitzmann et al. (2020).

Rigorous study of trigonometric networks and their connection to DGPs have been considered in Lu & Shafto (2022), and more general investigation of wide DNNs with bottlenecks in relation to DGPs have been considered in Agrawal et al. (2020); Pleiss & Cunningham (2021). Other closely related works include Meronen et al. (2020; 2021), who study calibration of shallow networks with periodic activations, Garnelo et al. (2018) proposes a different approach to combining aspects of GPs with DNNs, and the works Sun et al. (2020); Dutordoir et al. (2021) establish connections between neural network layers and inducing points for GPs/DGPs.

## 2 BACKGROUND

### 2.1 GAUSSIAN PROCESSES AND DEEP GAUSSIAN PROCESSES

A Gaussian process (GP) is a random function $f : \mathbb{R}^I \to \mathbb{R}$ such that for any $N > 0$ and any set of points $\boldsymbol{x}_n \in \mathbb{R}^I$ for $n = 1, \ldots, N$, we have that $(f(\boldsymbol{x}_1), \ldots, f(\boldsymbol{x}_N))^\top \in \mathbb{R}^N$ is Gaussian. GPs are characterised by a mean function $m : \mathbb{R}^I \to \mathbb{R}$ and a kernel $k : \mathbb{R}^I \times \mathbb{R}^I \to \mathbb{R}$, such that $\mathbb{E}[f(\boldsymbol{x})] = m(\boldsymbol{x})$ and $\mathrm{Cov}[f(\boldsymbol{x}), f(\boldsymbol{x}')] = k(\boldsymbol{x}, \boldsymbol{x}')$ for all $\boldsymbol{x}, \boldsymbol{x}' \in \mathbb{R}^I$ (Williams & Rasmussen, 2006). Extending these to have vector outputs $\boldsymbol{f} : \mathbb{R}^I \to \mathbb{R}^O$ is made possible by considering vector-valued means $\boldsymbol{m} : \mathbb{R}^I \to \mathbb{R}^O$ and matrix-valued kernels $\boldsymbol{k} : \mathbb{R}^I \times \mathbb{R}^I \to \mathbb{R}^{O \times O}$, satisfying $\mathbb{E}[f_i(\boldsymbol{x})] = m_i(\boldsymbol{x})$ and $\mathrm{Cov}[f_i(\boldsymbol{x}), f_i(\boldsymbol{x}')] = k_{ij}(\boldsymbol{x}, \boldsymbol{x}'), \forall i, j = 1, \ldots, O$. We write $\boldsymbol{f} \sim \mathcal{GP}(\boldsymbol{m}, \boldsymbol{k})$ to denote that $\boldsymbol{f}$ is a GP with mean $\boldsymbol{m}$ and kernel $\boldsymbol{k}$. A deep GP (DGP) $\boldsymbol{f} : \mathbb{R}^I \to \mathbb{R}^O$ extends GPs by considering compositions $\boldsymbol{f}(\boldsymbol{x}) = \boldsymbol{f}^L \circ \cdots \circ \boldsymbol{f}^1(\boldsymbol{x})$, where $\boldsymbol{f}^1 : \mathbb{R}^I \to \mathbb{R}^B$, $\boldsymbol{f}^\ell : \mathbb{R}^B \to \mathbb{R}^B$ for $\ell = 2, \ldots, L-1$ and $\boldsymbol{f}^L : \mathbb{R}^B \to \mathbb{R}^O$ are vector-GPs. The intermediate states $\mathbb{R}^B$ are referred to as the *bottlenecks*. We note that DGPs are more flexible class of models than GPs. However, due to their compositional structure, they are no longer Gaussian and therefore require approximate methods for inference, e.g. using variational Bayes.

### 2.2 RANDOM FOURIER FEATURES

Consider a zero-mean scalar GP $f \sim \mathcal{GP}(0, k)$ for some kernel $k$. We say that $k$ is *stationary* if there exists a function $\kappa : \mathbb{R}^I \to \mathbb{R}$ such that $k(\boldsymbol{x}, \boldsymbol{x}') = \kappa(\boldsymbol{x} - \boldsymbol{x}')$. In Rahimi & Recht (2007), it is shown that any stationary kernel on $\mathbb{R}^I$ can be expressed as an expectation

$$k(\boldsymbol{x}, \boldsymbol{x}') = 2\sigma^2 \mathbb{E}_{\boldsymbol{\omega}, b}\left[\cos(\boldsymbol{\omega}^\top \boldsymbol{x} + b)\cos(\boldsymbol{\omega}^\top \boldsymbol{x}' + b)\right] \tag{1}$$

$$\approx \frac{2\sigma^2}{H} \sum_{h=1}^H \cos(\boldsymbol{\omega}_h^\top \boldsymbol{x} + b_h)\cos(\boldsymbol{\omega}_h^\top \boldsymbol{x}' + b_h), \quad \boldsymbol{\omega}_h \sim p(\boldsymbol{\omega}), \quad b_h \sim U([0, 2\pi]) \tag{2}$$

for some $\sigma > 0$, where $p(\boldsymbol{\omega})$ is the normalised Fourier transform of the function $\kappa$ and $U([0, 2\pi])$ denotes the uniform distribution in the interval $[0, 2\pi]$. From the weight-space viewpoint of GPs, equation 2 implies that we have

$$f(\boldsymbol{x}) \approx \sum_{h=1}^H \theta_h \phi_h(\boldsymbol{x}), \quad \theta_h \sim \mathcal{N}(0, 1), \tag{3}$$

$$\text{where} \quad \phi_h(\boldsymbol{x}) = \sqrt{2\sigma^2/H}\cos(\boldsymbol{\omega}_h^\top \boldsymbol{x} + b_h), \quad h = 1, \ldots, H, \tag{4}$$

with $\boldsymbol{\omega}_h \sim p(\boldsymbol{\omega})$ and $b_h \sim U([0, 2\pi])$. Extension to vector-valued GPs $\boldsymbol{f} : \mathbb{R}^I \to \mathbb{R}^O$ with independent output components is straightforward, leading to a random Fourier feature representation of the form $\boldsymbol{f}(\boldsymbol{x}) = \boldsymbol{\Theta}\boldsymbol{\phi}(\boldsymbol{x})$ for $\boldsymbol{\Theta} \in \mathbb{R}^{O \times H}$ with $\Theta_{ij} \sim \mathcal{N}(0, 1)$, $i.i.d. \forall i, j$. For details and examples of random Fourier features, we refer the readers to Appendix A

## 3 DEEP RANDOM FEATURES FOR SPATIOTEMPORAL MODELLING

GPs are commonly used in spatiotemporal modelling due to their interpretability and smoothness inductive biases that are appealing to many geostatistical applications (Wikle et al., 2019). However, they are limited by their poor scalability and Gaussian assumptions. On the other hand, deep neural networks (DNN) offer a scalable, flexible modelling framework, however, do not have the desirable inductive bias of GPs. Motivated by this, we consider *deep random features* (Figure 2), which use random features corresponding to stationary GPs as building blocks for a larger neural network model tailored for spatiotemporal modelling, combining the benefits of both GPs and DNNs.

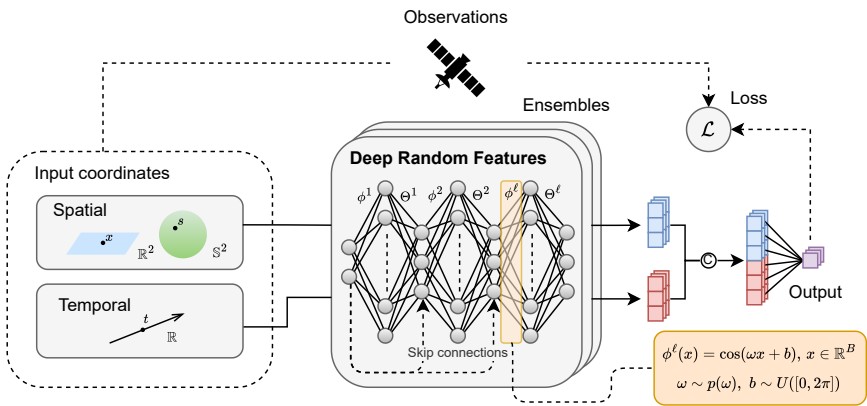

Figure 2: Illustration of spatiotemporal modelling with deep random features.

## 3.1 DEEP RANDOM FEATURES

In Section 2.2, we have seen that a shallow GP can be approximated by a combination of random features according to equation 3. In a similar fashion, Cutajar et al. (2017) propose a random feature expansion of deep GPs, by replacing each GP layer by their corresponding random features, yielding a DNN architecture that mimic the behaviour of the original DGP. In further details, let $\phi^1 : \mathbb{R}^I \to \mathbb{R}^H$ be a random feature (equation 4), which may be viewed equivalently as a single layer of a neural network with weights $\{\boldsymbol{\omega}_m\}_{m=1}^H$, biases $\{b_m\}_{m=1}^H$ and cosine activation. The first hidden layer in a deep GP, which itself is a vector-valued GP $\boldsymbol{f}^1 : \mathbb{R}^I \to \mathbb{R}^B$, can be approximated by a linear model

$$\boldsymbol{h}^1(\boldsymbol{x}) = \boldsymbol{\Theta}^1 \phi^1(\boldsymbol{x}), \tag{5}$$

where $\boldsymbol{\Theta}^1 \in \mathbb{R}^{B \times H}$ with $\Theta_{ij}^1 \sim \mathcal{N}(0,1)$, $i = 1, \ldots, B$, $j = 1, \ldots, H$. Similarly, given random features $\phi^\ell : \mathbb{R}^B \to \mathbb{R}^H$ for $\ell = 2, \ldots, L$, Gaussian weights $\boldsymbol{\Theta}^\ell \in \mathbb{R}^{B \times H}$ for $\ell = 2, \ldots, L-1$ and $\boldsymbol{\Theta}^L \in \mathbb{R}^{B \times O}$, we may consider a DNN $\boldsymbol{f}_{\boldsymbol{\Theta}} : \mathbb{R}^I \to \mathbb{R}^O$ of the form

$$\boldsymbol{f}_{\boldsymbol{\Theta}}(\boldsymbol{x}) = \boldsymbol{h}^L \circ \cdots \circ \boldsymbol{h}^2 \circ \boldsymbol{h}^1(\boldsymbol{x}), \quad \text{where} \quad \boldsymbol{h}^\ell(\boldsymbol{x}) := \boldsymbol{\Theta}^\ell \phi^\ell(\boldsymbol{x}), \quad \ell = 1, \ldots, L, \tag{6}$$

which we refer to as the random feature expansion of a DGP $\boldsymbol{f} : \mathbb{R}^I \to \mathbb{R}^O$. Generally, we may consider building DNNs independently of a DGP by using layers of the form $\boldsymbol{h}(\boldsymbol{x}) = \boldsymbol{\Theta}\phi(\boldsymbol{x})$ as building blocks for a neural network. We refer to such models as *deep random features*. Note that while each layer $\boldsymbol{h}^\ell(\boldsymbol{x})$ approximates a stationary GP, its compositions are no longer stationary.

### 3.1.1 TRAINING

In contrast to standard neural networks, when we train deep random features, we opt to alternate between trainable and fixed layers, where the parameters $\boldsymbol{\omega}_m^\ell, b_m^\ell$ in the layers $\phi_\ell(\cdot)$ are fixed upon initialisation, but the parameters $\boldsymbol{\Theta} := \{\boldsymbol{\Theta}^1, \ldots, \boldsymbol{\Theta}^L\}$ are trained. This is to mimic training of DGPs from the weight-space perspective, where inference should only be made with respect to $\boldsymbol{\Theta}$. Given a dataset $\mathcal{D} = \{(\boldsymbol{X}_n, \boldsymbol{y}_n)\}_{n=1}^N$, and an arbitrary loss $\ell : \mathbb{R}^O \times \mathbb{R}^O \to \mathbb{R}$, we minimise

$$\mathcal{L}_{\text{train}}(\boldsymbol{\Theta}; \mathcal{D}) = \frac{1}{N} \sum_{n=1}^N \ell(\boldsymbol{f}_{\boldsymbol{\Theta}}(\boldsymbol{X}_n), \boldsymbol{y}_n) + \beta \|\boldsymbol{\Theta}\|^2, \tag{7}$$

for some regularisation parameter $\beta > 0$. From a generalised Bayes' perspective, this is equivalent to maximum a priori estimation with the generalised posterior $p(\boldsymbol{\Theta}|\mathcal{D}) \propto \exp(-\ell(\boldsymbol{f}_{\boldsymbol{\Theta}}(\boldsymbol{X}), \boldsymbol{y}))p(\boldsymbol{\Theta})$ (Bissiri et al., 2016). Using mean-squared error as the loss and considering a shallow network, minimising equation 7 via gradient descent can be seen as sampling from the GP posterior in the neural tangent kernel limit (Lee et al., 2019; He et al., 2020). Using other losses such as the Huber loss, this becomes akin to robust GP regression (Algikar & Mili, 2023; Altamirano et al., 2024).

### 3.1.2 SPHERICAL RANDOM FEATURES

When modelling signals over the sphere, which arises when we need to interpolate global satellite measurements (see Figure 1), we require an analogous notion of random features defined over the sphere. In Borovitskiy et al. (2020), commonly used kernels such as the Matérn kernels are extended to be defined over general Riemannian manifolds, including the two-sphere $\mathbb{S}^2$. In general, such kernels can be approximated by the Mercer sum

$$k(s, s') \approx \frac{1}{C_\Phi} \sum_{j=0}^{J} \Phi(\lambda_j) \varphi_j(s) \varphi_j(s'), \quad s, s' \in \mathbb{S}^2, \tag{8}$$

for some $\Phi : \mathbb{R} \to \mathbb{R}$ and constant $C_\Phi$ determined from the kernel, and $\{\lambda_j\}_{j=0}^{J}$, $\{\varphi_j\}_{j=0}^{J}$ are the $J$ top eigenvalues and eigenfunctions respectively of the negative Laplace-Beltrami operator; on $\mathbb{S}^2$, the latter is precisely the spherical harmonics. Furthermore, on $\mathbb{S}^2$, by making use of the addition theorem for spherical harmonics and the result (Azangulov et al., 2024, Proposition 7), we get an alternative expression for the kernel (see Appendix A.2 for the derivation)

$$k(s, s') \approx \mathbb{E}_{\omega, b} \left[ c_\omega \, \mathcal{G}_\omega^{1/2}(d_{\mathbb{S}^2}(s, b)) \, \mathcal{G}_\omega^{1/2}(d_{\mathbb{S}^2}(s', b)) \right], \quad s, s' \in \mathbb{S}^2, \tag{9}$$

where $\mathcal{G}_n^\alpha(\cdot)$ are the Gegenbauer polynomials of order $n$ and weight parameter $\alpha$, $d_{\mathbb{S}^2}(\cdot, \cdot)$ denotes the geodesic distance on $\mathbb{S}^2$, $c_\omega$ is an appropriate scaling constant, and the expectation is taken over $b \sim U(\mathbb{S}^2)$, the uniform distribution over the sphere, and $\omega \sim \text{Multinomial}(C_\Phi^{-1}\Phi(\lambda_1), \ldots, C_\Phi^{-1}\Phi(\lambda_J))$. Then, by considering Monte Carlo approximation of the expectation in equation 9, this implies random feature maps of the form

$$\phi_{\mathbb{S}^2}^m(s) = \sqrt{M^{-1}c_{\omega_m}} \, \mathcal{G}_{\omega_m}^{1/2}(d_{\mathbb{S}^2}(s, b_m)), \quad s \in \mathbb{S}^2, \quad m = 1, \ldots, M, \tag{10}$$

$$\text{where} \quad \omega_m \sim \text{Multinomial}(C_\Phi^{-1}\Phi(\lambda_1), \ldots, C_\Phi^{-1}\Phi(\lambda_J)), \quad b_m \sim U(\mathbb{S}^2). \tag{11}$$

This gives us an analogous notion of random features (equation 4) on the sphere, which we can use as a component in our deep random feature model when our input is spherical.

**Remark 1** *We may also consider the deterministic features $\phi^m(s) = \sqrt{C_\Phi^{-1}\Phi(\lambda_m)}\varphi_m(s)$, derived from equation 8, which is analogous to the regular Fourier features (Hensman et al., 2018; Solin & Särkkä, 2020) in the planar case. However in practice, we find that working with random features (equation 10) produce more stable results when using single precision arithmetic.*

### 3.1.3 SPATIOTEMPORAL MODELLING WITH DEEP RANDOM FEATURES

So far, we have only discussed how to process spatial inputs. In order to deal with the temporal components in our data, we first consider deep random features in the spatial domain $\boldsymbol{h}_x^{(L_x)} : \mathcal{X} \to \mathbb{R}^B$ ($\mathcal{X} = \mathbb{R}^I$ or $\mathbb{S}^2$), and temporal domain $\boldsymbol{h}_t^{(L_t)} : \mathbb{R} \to \mathbb{R}^B$ separately, before combining them as

$$\boldsymbol{f}(x, t) = \boldsymbol{\Theta}(\text{concat}[\boldsymbol{h}_x^{(L_x)}(x), \boldsymbol{h}_t^{(L_t)}(t)]), \tag{12}$$

where $\boldsymbol{\Theta} \in \mathbb{R}^{O \times 2B}$ are learnable weights initialised with i.i.d. standard Gaussians. At short timescales, geospatial fields are approximately stationary, hence we can use a single layer network to model the temporal component $\boldsymbol{h}_t^{(L_t)}$ (i.e., $L_t = 1$). To introduce more complex spatiotemporal dependence, we can replace the linear output layer in equation 12 with deep random features. However we find that in most applications this is unnecessary, only introducing extra cost.

### 3.1.4 SKIP CONNECTIONS

To prevent pathological behaviour from emerging as we increase the network depth, we add skip connections to the inputs (Duvenaud et al., 2014; Dunlop et al., 2018). In the planar case, this takes

$$\boldsymbol{h}^{(\ell+1)}(\boldsymbol{x}) = \boldsymbol{\Theta}^{\ell+1} \boldsymbol{\phi}^{\ell+1}(\text{concat}[\boldsymbol{h}^{(\ell)}(\boldsymbol{x}), \boldsymbol{x}]) \tag{13}$$

in the $(\ell + 1)$-th layer of the network, where $\phi^{\ell+1} : \mathbb{R}^{B+I} \to \mathbb{R}^M$. In the spherical case, this is not straightforward as the outputs of each layer will be Euclidean while the input is spherical. To this end, we consider $\phi^{\ell+1}$ to be additive random features corresponding to the sum kernel $\boldsymbol{k}((\boldsymbol{x}, s), (\boldsymbol{x}', s')) = \boldsymbol{k}_{\mathbb{R}^B}(\boldsymbol{x}, \boldsymbol{x}') + \boldsymbol{k}_{\mathbb{S}^2}(s, s')$, for $\boldsymbol{x}, \boldsymbol{x}' \in \mathbb{R}^B$, $s, s' \in \mathbb{S}^2$, where $k_{\mathbb{R}^B}(\cdot, \cdot)$, $k_{\mathbb{S}^2}(\cdot, \cdot)$ are stationary kernels on their respective spaces (see Appendix B.1 for details).

## 3.2 Uncertainty quantification

Uncertainty quantification (UQ) with deep random features is achieved using standard Bayesian deep learning techniques. In particular, we consider the following methods in our experiments:

**Variational inference.** This considers a Gaussian approximation to the posterior $p(\boldsymbol{\Theta}|\mathcal{D}) \approx \mathcal{N}(\boldsymbol{\Theta}|\boldsymbol{m}, \boldsymbol{C})$. Here, the moments of the variational distribution $q(\boldsymbol{\Theta}) := \mathcal{N}(\boldsymbol{\Theta}|\boldsymbol{m}, \boldsymbol{C})$ are learned by maximising the evidence lower bound (ELBO)

$$\mathcal{L}_{\mathrm{ELBO}}(\boldsymbol{f}_{\boldsymbol{\Theta}}; \mathcal{D}) = \mathbb{E}_q\left[-\ell(\boldsymbol{f}_{\boldsymbol{\Theta}}(\boldsymbol{X}), \boldsymbol{y})\right] - \mathcal{KL}(q||p_{\boldsymbol{\Theta}}), \tag{14}$$

where $p_{\boldsymbol{\Theta}}(\boldsymbol{\Theta}) = \mathcal{N}(\boldsymbol{\Theta}|\boldsymbol{0}, \boldsymbol{I})$ is the prior on $\boldsymbol{\Theta}$ and $\mathcal{KL}(\cdot||\cdot)$ is the Kullback-Leibler divergence.

**Dropout at test time.** A simple heuristic for obtaining uncertainty estimates is to apply dropout not only at training time but also at test time. This yields an ensemble of random outputs, whose empirical distribution informs us of the model uncertainty. In fact, one can understand this as a form of variational inference, as shown in Gal & Ghahramani (2016).

**Deep ensembles.** Deep ensembles (Lakshminarayanan et al., 2017) obtain uncertainty estimates by training an ensemble of models, initialised from different random seeds. The ensemble of outputs is then used to estimate uncertainty, similar to the dropout method for UQ. While being a simple method, this has been shown to be surprisingly effective at obtaining uncertainty estimates. Moreover, provided the model is small enough (which is often the case for deep random features), the ensembles can be trained in parallel on a single GPU.

## 3.3 Hyperparameter selection

Our deep random features model contain several hyperparameters $\boldsymbol{\lambda}$, including those of the kernel (e.g. lengthscales) that we use to derive our random features. If variational inference is used for UQ, we can take the ELBO (equation 14) for model comparison, as it may be viewed as a surrogate for the log model evidence $\log p(\mathcal{D}|\boldsymbol{\lambda})$, being its lower bound. When using ensemble based methods, we rely on performance on a held-out validation set $\mathcal{D}^* = \{(\boldsymbol{x}_n^*, \boldsymbol{y}_n^*)\}_{n=1}^{N^*}$ to select our hyperparameters. In particular, we consider the following validation loss to select $\boldsymbol{\lambda}$

$$\mathcal{L}_{\mathrm{val}}(\boldsymbol{\lambda}) = \frac{1}{N^*}\sum_{n=1}^{N^*}\frac{1}{J}\sum_{j=1}^{J}\ell(\boldsymbol{f}_j(\boldsymbol{x}_n^*; \mathcal{D}, \boldsymbol{\lambda}); \boldsymbol{y}_n^*), \tag{15}$$

where $\{\boldsymbol{f}_j\}_{j=1}^{J}$ are the ensembles. Note that this can be viewed as approximating the negative log-predictive density (see Remark 3, Appendix C.1). We minimise this loss using Bayesian optimisation. In practice, we find that learning $\boldsymbol{\lambda}$ from equation 15 alone may still lead to overfitting models. Therefore to prevent this, we may opt to add an extra *functional regularisation* term

$$\mathcal{L}_{\mathrm{val+reg}}(\boldsymbol{\lambda}) = (1-\alpha)\mathcal{L}_{\mathrm{val}}(\boldsymbol{\lambda}) + \alpha\|\nabla\bar{\boldsymbol{f}}(\cdot; \mathcal{D}, \boldsymbol{\lambda})\|_{L^2}^2, \quad \bar{\boldsymbol{f}} := \frac{1}{J}\sum_{j=1}^{J}\boldsymbol{f}_j \tag{16}$$

for $\alpha \in [0, 1)$, which helps to penalise those $\boldsymbol{\lambda}$ that give rise to functions $\boldsymbol{f}$ with sharp gradients (see Remark 2 in Appendix B.2 on why we do not consider hyperpriors for regularisation). Here, $\|\cdot\|_{L^2}^2$ denotes the appropriate $L^2$ norm depending on the input space and $\nabla$ the gradient. For spherical inputs, we refer the readers to Appendix B.2 for more details.

## 4 Experiments

We evaluate the spatiotemporal deep random features (DRF) model on various remote sensing datasets and compare against various baselines to assess its ability to make predictions and quantify uncertainty. In our first experiment, we consider interpolation of synthetic data, and evaluate our model's ability to recover the ground truth. In our second and third experiments, we consider interpolation of real satellite data at local and global scales to test the robustness of our method. Details can be found in Appendix C. All experiments are performed using the NVIDIA L4 GPU.

## 4.1 BASELINE MODELS

Throughout this section, we consider several models as baselines to compare our model against. We consider both GP-based baselines and DNN-based baselines. In the former category, we consider the sparse variational GP (SVGP) model in Hensman et al. (2013), deep GPs (DGP) using doubly stochastic variational inference (Salimbeni & Deisenroth, 2017) and a mixture model of local GPs using the `GPSat` library (Gregory et al., 2024b). In the latter caregory, we consider deep ensembles of multilayer perceptrons (MLP) with ReLU activations, Fourier features network (FFN, Tancik et al. (2020)), which use random Fourier features in the first layer only, MLP with sinusoidal activations (SIREN, Sitzmann et al. (2020)) and conditional neural processes Garnelo et al. (2018).

## 4.2 EVALUATION ON SYNTHETIC DATA

The purpose of our first experiment is to use synthetic observations from a ground truth field to evaluate our model's ability to reconstruct the field. We use mean sea surface height (MSS) in the arctic as our ground truth, synthesised from 12 years of altimetry readings of Sentinel-3A, 3B (S3A, 3B) and CryoSat-2 (CS2) satellites. We then generate artificial measurements along S3A, 3B and CS2 tracks between the dates March 1st–10th 2020, taking the MSS values along the tracks and adding i.i.d. Gaussian noise to mimic measurement noise. Our final dataset comprise 1,158,505 datapoints; we select 80% of these randomly for training and the remaining 20% for validation. We train all models using the mean-squared error loss (for GP baselines, this corresponds to a Gaussian likelihood) with fixed weight decay parameter matching the observation noise variance. Visual comparison of predictions from all models can be found in Appendix C.5.1.

### 4.2.1 EFFECT OF DEPTH

In Figure 3, we show the effect of depth on our model's ability to reconstruct the true MSS field (in terms of RMSE) and corresponding computation time of the entire workflow, including the time to tune the kernel hyperparameters (see Appendix C.2.2). Generally, we find that deep networks outperform the shallow network on the RMSE, with the four layer model performing the best on this example. With $> 4$ layers, we start to see some overfitting, which explains the higher RMSE for the 10 and 20 layer models. In Figure 4, we display mean results for models with two and four layers. We see that the deeper network is able to capture higher frequency details, resulting in the improved RMSE. The time it takes to train and tune models with 1-4 layers are not significantly different.

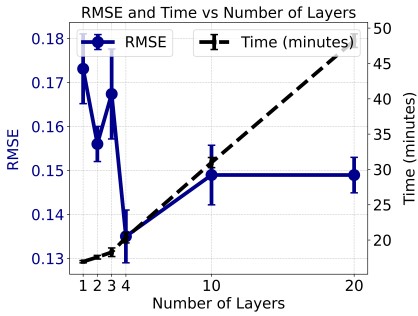

Figure 3: Comparison of RMSE and computation time vs. number of layers.

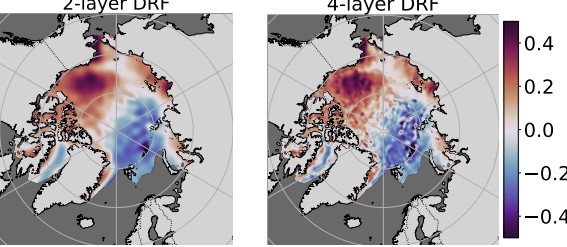

Figure 4: Comparison of predictions from DRF with two layers (left) and four layers (right). The four layer model is able to capture finer details compared to the two layer model.

### 4.2.2 UQ COMPARISONS

Next, we compare various UQ methods applied to a DRF model with four layers. In Table 1, we display comparisons with respect to the root mean squared error (RMSE), the negative log-likelihood (NLL), and the continuous ranked probability score (CRPS) (see Appendix C.1 for details on our evaluation method). The latter two evaluate the quality of uncertainties produced. Overall, we find that deep ensembles produce the best result in the CRPS and the RMSE, whereas variational inference (VI) yielded the best NLL. The lower NLL using VI may be due to the fact that its predictions are typically underconfident (see Figure 8, Appendix C.5.1) and NLL penalises them more lightly

| Model | CRPS | NLL | RMSE | Time (minutes) |
|---|---|---|---|---|
| DRF (Ensembles) | **0.046 ± 0.005** | 13.590 ± 4.899 | 0.135 ± 0.006 | 19.7 ± 0.4 |
| DRF (VI) | 0.071 ± 0.019 | **−0.407 ± 0.756** | 0.166 ± 0.021 | 6.40 ± 0.06 |
| DRF (Dropout) | 0.174 ± 0.001 | 425.987 ± 208.969 | 0.238 ± 0.001 | 48.6 ± 2.5 |
| SVGP | 0.230 ± 0.001 | 320.811 ± 52.960 | 0.155 ± 0.002 | 14.6 ± 0.0 |
| DGP | 0.058 ± 0.001 | 1614.069 ± 328.517 | 0.135 ± 0.002 | 42.3 ± 0.03 |
| GPSat | **0.045 ± 0.007** | 74.738 ± 15.622 | **0.126 ± 0.001** | 63.6 ± 0.2 |
| ReLU MLP | 0.062 ± 0.000 | 30.504 ± 7.877 | 0.146 ± 0.000 | 10.07 ± 0.01 |
| FFN | 0.072 ± 0.008 | 126.869 ± 111.178 | 0.153 ± 0.005 | 31.3 ± 0.3 |
| SIREN | 0.066 ± 0.000 | 13.974 ± 0.393 | 0.155 ± 0.000 | 2.55 ± 0.002 |
| CNP | 0.238 ± 0.070 | 2.525 ± 0.459 | 0.202 ± 0.010 | 21.0 ± 0.1 |

Table 1: Comparison of the CRPS, NLL and RMSE scores for a four-layer DRF (with different UQ methods) against various baselines on the synthetic experiment. Best performing model in **bold**, second best performing in blue and third best performing in orange. We display the mean and standard deviation over five experiments.

than overconfident ones. However, the results in Figure 8 suggest that the results from deep ensembles are better calibrated to the observations, which explains the lower CRPS. Dropout does not perform well in neither the mean prediction nor uncertainty estimation.

### 4.2.3 BASELINE COMPARISONS

In Table 1, we also display results for the other baseline models described in Section 4.1. Regarding computation times, for DRF and FFN, we include the time to tune the kernel hyperparameters using Bayesian optimisation (Section 3.3) to make a fair comparison with the GP-based baselines, where the total time for training and inference are recorded. However, we assume other hyperparameters, such as number of layers and hidden units to be fixed ($L = 4$, $B = 128$, $H = 1000$). For the other DNN-based baselines, we assume the architecture is tuned ahead of time and fixed.

Comparing with the GP baselines, we find that DGP and the GPSat model to be closest competitors to the DRF deep ensembles, with GPSat surpassing its performance on the RMSE. However, the time taken to train the DGP and GPSat model are two to three times longer than the time taken to train and tune the ensemble DRF. For example, GPSat trains 1225 local GP models on this example, which makes computation heavy. DGP has overall low predictive variances (see Figure 9, Appendix C.5.1), which results in high NLL values. In contrast, the uncertainty estimates of DRF and GPSat are well-calibrated to the satellite tracks. Qualitatively, all three models recover the ground truth well, with GPSat and DRF reconstructing it almost perfectly.

Comparing to other DNN baselines, SIREN's performance is noteworthy, being similar to DRF in that it uses trigonometric activations and differing only in the way the weights are initialised and whether it has bottleneck layers with fixed preactivations. The DRF ensemble is better able to capture the spatiotemporal patterns of the field, reinforcing the importance of the subtle architectural differences. The ReLU network, FFN and CNP all lead to oversmoothing results, similar to SVGP.

### 4.3 FREEBOARD ESTIMATION FROM SATELLITE ALTIMETRY DATA

Here, we consider the interpolation of real altimeter readings of sea-ice freeboard taken along the Sentinel-3A, 3B and CryoSat-2 satellites (Gregory et al., 2024a). Real satellite altimetry measurements are typically noisy with heavy-tailed statistics (see Figure 7, Appendix C.3), hence they present a more challenging setting than our previous synthetic experiment. Our goal here is to test the robustness of our approach in comparison to other methods in this more realistic setting. Experimental details can be found in Appendix C.3 and visual comparison of all results can be found in Figure 10, Appendix C.5.2.

We compare a two-layer DRF against the same baselines as before and display the root mean absolute error (RMAE), CRPS and negative log-predictive density (NLPD) on a separately held out test data comprising 15% of the entire data in Table 2. We use the RMAE instead of the RMSE here

as it is more robust to the heavy-tailed statistics of the measurement error, and therefore provides a more reliable performance metric in this setting. The hyperparameter search for DRF was performed with functional regularisation (see Section 3.3) as we found that without it, optimising on only the validation loss lead to overfitting models (see Figure 5). Here, we used a penalty weight of $\alpha = 0.9$.

We find that out of our GP-based baselines, SVGP and the `GPSat` model give comparable performance to DRF ensembles, with `GPSat` giving the best performance quantitatively. However, when we examine the outputs from all three models (Figure 5), we find that the results from `GPSat` contain spurious patterns resulting from unstable hyperparameter optimisation at several local expert locations. This issue occurs since the `GPSat` local experts only see data in local regions, making them more sensitive to the heavy noise present in the data. On the other hand, DRF sees data globally, which helps them to identify the larger structures in the data, while simultaneously capturing the finer details owing to their deep architecture. We find that qualitatively, SVGP performs well on this example, due to the larger prominence of low frequency features in the underlying field that extend across the basin. We see that DRF provides a middle ground between the two, being neither "too local" as we see in `GPSat`, nor "too global", demonstrating its ability to adapt to the characteristics of the field. The other DNN baselines are found to perform poorly, with the ReLU MLP and CNP showing especially poor fit. The quantitative metrics for FFN and SIREN are actually decent, with FFN performing best on RMAE. However, the qualitative results in Figure 10 show signs of heavy overfitting with both models, which is especially prominent in SIREN.

| Model | CRPS | NLPD | RMAE |
|---|---|---|---|
| DRF (Ensembles) | $\mathbf{0.077 \pm 0.000}$ | $-0.944 \pm 0.020$ | $0.322 \pm 0.000$ |
| SVGP | $0.079 \pm 0.001$ | $\mathbf{-1.300 \pm 0.004}$ | $0.322 \pm 0.001$ |
| DGP | $0.208 \pm 0.000$ | $-0.159 \pm 0.000$ | $0.339 \pm 0.001$ |
| GPSat | $\mathbf{0.076 \pm 0.001}$ | $-1.167 \pm 0.025$ | $0.318 \pm 0.000$ |
| ReLU MLP | $0.714 \pm 0.936$ | $0.076 \pm 1.578$ | $0.751 \pm 0.554$ |
| FFN | $0.080 \pm 0.001$ | $2.145 \pm 1.811$ | $\mathbf{0.316 \pm 0.004}$ |
| SIREN | $0.088 \pm 0.000$ | $-1.109 \pm 0.010$ | $0.345 \pm 0.001$ |
| CNP | $0.101 \pm 0.001$ | $-0.898 \pm 0.030$ | $0.328 \pm 0.002$ |

Table 2: Comparison of the CRPS, NLPD and RMAE scores for a two-layer DRF against various baselines on sea-ice freeboard interpolation from S3A, 3B and CS2 satellite altimetry readings. Best performing model in **bold**, second best performing in blue and third best performing in orange.

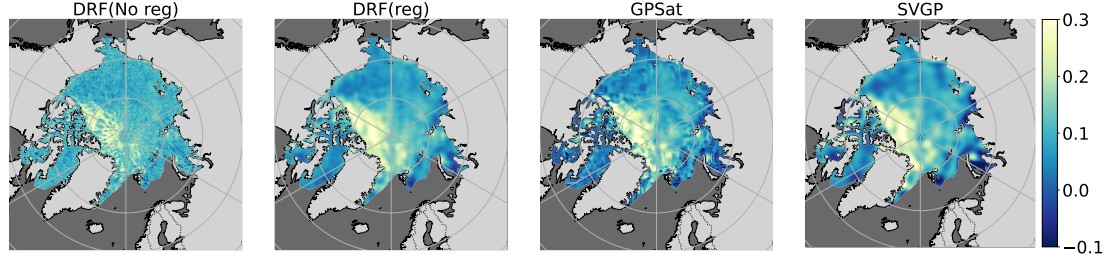

Figure 5: Mean results of DRF, GPSat and SVGP on freeboard interpolation. For DRF, we plot results obtained both with and without functional regularisation during hyperparameter search.

## 4.4 LARGE SCALE INTERPOLATION OF GLOBAL SEA LEVEL ANOMALY

In this final experiment, we investigate the potential of DRF to interpolate *global* fields using spherical random features (Section 3.1.2) in the spatial inputs. For this experiment, we use real data of sea level anomaly measurements collected from the Sentinel 3A, 3B satellites (Copernicus Data Space Ecosystem, 2024). By considering four days of measurements, our final data consists of 8,094,569 datapoints. We use 80% for training, and 20% for validation. Similar to our previous data obtained from real satellite measurements, this data contains many outliers, making it a challenge to interpolate the data robustly, let alone whilst being consistent with the geometry of the sphere.

Our goal is to fit a spatiotemporal field $f : \mathbb{S}^2 \times \mathbb{R} \to \mathbb{R}$. To this end, we consider a DRF model whose first layer in the spatial component is given by the spherical random feature $\phi_{\mathbb{S}^2} : \mathbb{S}^2 \to \mathbb{R}^H$ (equation 10). The subsequent layers are given by Euclidean random feautures. To train our model, we opted to use the Huber loss instead of MSE, which gave rise to slightly more robust results, likely due to the large number of outliers. We also used functional regularisation with a penalty weight of $\alpha = 0.95$ when tuning hyperparameters.

In Figure 6, we compare the mean predictions of the spherical DRF model with predictions from (1) SVGP using the spherical Matérn kernel of Borovitskiy et al. (2020), and (2) the Euclidean DRF model, taking the longitude and latitude coordinates of the satellite tracks as spatial inputs in $\mathbb{R}^2$. We use the spherical Matérn kernel implementation in the `geometric-kernels` package Mostowsky et al. (2024) to model the spatial component of our SVGP baseline. The temporal component is included by modelling the GP with a product kernel $k((x,t),(x',t')) = k_{\mathbb{S}^2}(x, x')k_{\mathbb{R}}(t, t')$. Comparing the SVGP output with DRF, we see that they are both able to capture the larger patterns in the data. However, SVGP fails to capture some of the finer fluctuations (as also indicated by quantitative metrics in Table 3 in Appendix C.4.3), for instance those around the Antarctic circumpolar current, known for its intense ocean activities.

For the Euclidean DRF, while it admits a deep structure, we find that it is not flexible enough to adapt to the spherical geometry of the input space. For example, there are spurious distortions around the poles cause by the stereographic projection, in addition to a discontinuity at longitude $= 0°$ (see Figure 11 in Appendix C.5.3). Perhaps more interestingly, the Euclidean DRF is not able to learn the fine scale fluctuations that the spherical DRF is able to pick up, only being able to learn large scale trends in the data, similar to SVGP. This highlights the importance of explicitly incorporating the spherical inductive bias into the model when modelling global fields.

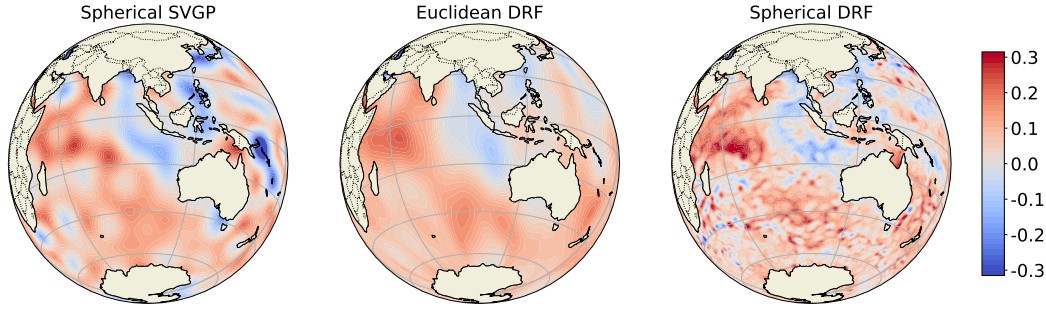

Figure 6: Mean results for global sea level anomaly interpolation. From left to right: SVGP using the spherical Matérn kernel, Euclidean DRF, and Spherical DRF. Spherical DRF is able to learn more intricate details compared to the other two baselines.

## 5 CONCLUSION

In this paper, we propose to model spatiotemporal fields using deep neural networks, whose layers are derived from random feature expansions of stationary kernels. This neural representation can be trained on observations to effectively fill in the gaps between remote sensing observations of the earth's surface. Our experiments on various remote sensing data demonstrate that the deep ensemble model is able to flexibly adapt to the data, being able to learn both low and high-frequency structures that exist in the underlying field. A current limitation of our approach is the difficulty of tuning kernel hyperparameters; we use Bayesian optimisation (BO) on the validation loss to achieve this, which require knowledge of the ranges each hyperparameter may take. This is not clear due to the deep architecture, making the hyperparameters less interpretable than the shallow case. Additionally, we observe that it is sometimes necessary to add functional regularisation to reduce BO variance, necessitating hand tuning of the penalty weight $\alpha$, a hyper-hyperparameter. Despite this, our promising results suggest the potential for deep learning methods to pave the way for more accurate and flexible reconstructions of spatiotemporal fields from remote sensing data.

ACKNOWLEDGMENTS

WC and MT acknowledge support from ESA (Clev2er: CRISTAL LEVel-2 procEssor prototype and R&D); MT acknowledges support from (#ESA/AO/1-9132/17/NL/MP, #ESA/AO/1-10061/19/I-EF, SIN'XS: Sea Ice and Iceberg and Sea-ice Thickness Products Inter-comparison Exercise) and NERC (#NE/T000546/1 761 and #NE/X004643/1). AM is supported by the EPSRC-funded UCL Centre for Doctoral Training in Intelligent, Integrated Imaging in Healthcare (i4health) (EP/S021930/1). ST is supported by a Department of Defense Vannevar Bush Faculty Fellowship held by Prof. Andrew Stuart, and by the SciAI Center, funded by the Office of Naval Research (ONR), under Grant Number N00014-23-1-2729. We also thank Ronald MacEachern for generating and sharing the MSS data, William Gregory for support on the GPSat baseline experiments, Daniel Augusto de Souza and Marc Peter Deisenroth for useful initial discussions, Viacheslav Borovitskiy for tutoring ST about spherical feature maps, Michalis Michaelides for useful feedbacks on the first draft of our manuscript, and PhysicsX for providing additional GPU support.

CODE AVAILABILITY

The code for reproducing our experiments is available in the following github repository

`https://github.com/totony4real/DeepRandomFeatures`.

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

# A  RANDOM FEATURES FOR STATIONARY GAUSSIAN PROCESSES

## A.1  RANDOM FOURIER FEATURES FOR PLANAR GAUSSIAN PROCESSES

Let $k : \mathbb{R}^d \times \mathbb{R}^d \to \mathbb{R}$ be a stationary kernel. That is, there exists a function $\kappa : \mathbb{R}^d \to \mathbb{R}$ such that $k(\boldsymbol{x}, \boldsymbol{x}') = \kappa(\boldsymbol{x} - \boldsymbol{x}')$. Then Bochner's theorem states that there exists a spectral density $s : \mathbb{R}^d \to \mathbb{R}$, such that

$$\kappa(\boldsymbol{x} - \boldsymbol{x}') = \int_{\mathbb{R}^d} s(\boldsymbol{\omega}) e^{i\boldsymbol{\omega}^\top (\boldsymbol{x} - \boldsymbol{x}')} \tag{17}$$

$$= \sigma^2 \mathbb{E}_{\boldsymbol{\omega} \sim p(\boldsymbol{\omega})} \left[ \zeta_\omega(\boldsymbol{x}) \zeta_\omega^*(\boldsymbol{x}') \right], \tag{18}$$

where $\zeta_\omega(\boldsymbol{x}) := e^{i\boldsymbol{\omega}^\top \boldsymbol{x}}$, $\sigma^2 := \int_{\mathbb{R}^d} s(\boldsymbol{\omega}) \mathrm{d}\boldsymbol{\omega}$ and $p(\boldsymbol{\omega}) = s(\boldsymbol{\omega})/\sigma^2$, so that $p(\boldsymbol{\omega})$ is a probability density function. Expanding this further, we have

$$18 = \sigma^2 \mathbb{E}_{\boldsymbol{\omega} \sim p(\boldsymbol{\omega})} [\cos(\boldsymbol{\omega}^\top \boldsymbol{x}) \cos(\boldsymbol{\omega} \cdot \boldsymbol{x}') + \sin(\boldsymbol{\omega}^\top \boldsymbol{x}) \sin(\boldsymbol{\omega}^\top \boldsymbol{x}')] \tag{19}$$

$$- i\sigma^2 \underbrace{\mathbb{E}_{\boldsymbol{\omega} \sim p(\boldsymbol{\omega})} [\cos(\boldsymbol{\omega}^\top \boldsymbol{x}) \sin(\boldsymbol{\omega}^\top \boldsymbol{x}') + \sin(\boldsymbol{\omega}^\top \boldsymbol{x}) \cos(\boldsymbol{\omega}^\top \boldsymbol{x}')]}_{=0 \text{ (since function is odd)}} \tag{20}$$

$$= \sigma^2 \mathbb{E}_{\boldsymbol{\omega} \sim p(\boldsymbol{\omega})} \left[ \frac{1}{\pi} \int_0^{2\pi} \cos(\boldsymbol{\omega}^\top \boldsymbol{x} + b) \cos(\boldsymbol{\omega}^\top \boldsymbol{x}' + b) \mathrm{d}b \right] \tag{21}$$

$$= 2\sigma^2 \mathbb{E}_{\boldsymbol{\omega} \sim p(\boldsymbol{\omega})} \left[ \mathbb{E}_{b \sim U[0,2\pi]} \left[ \cos(\boldsymbol{\omega}^\top \boldsymbol{x} + b) \cos(\boldsymbol{\omega}^\top \boldsymbol{x}' + b) \right] \right] \tag{22}$$

$$\approx \frac{2\sigma^2}{N} \sum_{m=1}^M \cos(\boldsymbol{\omega}_m^\top \boldsymbol{x} + b_n) \cos(\boldsymbol{\omega}_n^\top \boldsymbol{x}' + b_m), \tag{23}$$

$$\text{where } \boldsymbol{\omega}_n \sim p(\boldsymbol{\omega}), \quad b_n \sim U[0, 2\pi]. \tag{24}$$

The final expression gives us the random features

$$\phi^m(\boldsymbol{x}) = \sqrt{\frac{2\sigma^2}{M}} \cos(\boldsymbol{\omega}_m^\top \boldsymbol{x} + b_m), \quad m = 1, \ldots, M, \tag{25}$$

corresponding to the stationary kernel $k$. The only information that changes as we change the kernel is the (normalised) spectral density $p(\boldsymbol{\omega})$, used to sample the weights $\boldsymbol{\omega}$. Below, we give examples of such $p(\boldsymbol{\omega})$ for the squared-exponential and Matérn kernels.

**Example 1 (Squared-exponential kernel)** *The squared-exponential kernel is given by*

$$\kappa(\boldsymbol{x} - \boldsymbol{x}') = \sigma^2 \exp\left(-\frac{\|\boldsymbol{x} - \boldsymbol{x}'\|^2}{2\ell^2}\right), \tag{26}$$

*where $\ell$ is the lengthscale hyperparameter and $\sigma^2$ is the kernel variance. The Fourier transform of $\kappa$ (up to a normalisation constant) can be checked to be given by*

$$p(\boldsymbol{\omega}) = \frac{\ell^d}{(2\pi)^{d/2}} \exp\left(-\frac{\|\boldsymbol{\omega}\|^2 \ell^2}{2}\right). \tag{27}$$

*Note that this is precisely the probability density function of a multivariate Gaussian $\mathcal{N}(\boldsymbol{0}, \ell^{-2}\boldsymbol{I})$. Thus, we obtain a random features approximation to the squared-exponential GP by sampling $\boldsymbol{\omega}_m \sim \mathcal{N}(\boldsymbol{0}, \ell^{-2}\boldsymbol{I})$ in equation 24.*

**Example 2 (Matérn kernel)** *The Matérn kernel is given by*

$$\kappa(\boldsymbol{x} - \boldsymbol{x}') = \alpha^2 \frac{2^{1-\nu}}{\Gamma(\nu)} \left(\sqrt{2\nu} \frac{\|\boldsymbol{x} - \boldsymbol{x}'\|}{\ell}\right)^\nu K_\nu\left(\sqrt{2\nu} \frac{\|\boldsymbol{x} - \boldsymbol{x}'\|}{\ell}\right), \tag{28}$$

where $\nu$ is the smoothness hyperparameter and $K_\nu$ is the modified Bessel function of the second kind. It is well-known that the Fourier transform of the Matérn kernel equation 28 reads

$$p(\omega) = \frac{2^d \pi^{d/2} \Gamma(\nu + \frac{d}{2})(2\nu)^\nu}{\Gamma(\nu)\ell^{2\nu}} \left( \frac{2\nu}{\ell^2} + 4\pi^2 \omega^2 \right)^{-\nu + \frac{d}{2}}, \tag{29}$$

and in fact, this can be identified as the probability density function of the multivariate t-distribution $t_{2\nu}(\mathbf{0}, \ell^{-2}\mathbf{I})$. Thus, we obtain a random features approximation to the Matérn GP by sampling $\boldsymbol{\omega}_m \sim t_{2\nu}(\mathbf{0}, \ell^{-2}\mathbf{I})$ in equation 24.

## A.2 Matérn Gaussian processes on the sphere

Following Borovitskiy et al. (2020), we define a Matérn Gaussian process on the two-sphere $\mathbb{S}^2$ with lengthscale $\ell$ and smoothness parameter $\nu$ to be a stochastic process $f$ defined by the solution to the stochastic partial differential equation

$$\left( \frac{2\nu}{\ell^2} - \Delta_{\mathbb{S}^2} \right)^{\frac{\nu+1}{2}} f = \dot{\mathcal{W}}_\sigma. \tag{30}$$

Here, $\dot{\mathcal{W}}_\sigma$ denotes the space-time white noise process over $L^2(\mathbb{S}^2)$ with spectral density $\sigma$ and $\Delta_{\mathbb{S}^2}$ denotes the Laplace-Beltrami operator on the two-sphere (see Borovitskiy et al. (2020) for the notion of solution to this equation). By (Borovitskiy et al., 2020, Theorem 5), the corresponding kernel has an explicit expression of the form (follows from Mercer's theorem)

$$k_{\mathbb{S}^2}(s, s') = \frac{\sigma^2}{C_\nu} \sum_{j=0}^{\infty} \sum_{k=1}^{2j+1} \left( \frac{2\nu}{\ell^2} + \lambda_j \right)^{-\nu-1} \varphi_{j,k}(s)\varphi_{j,k}(s'), \quad \forall s, s' \in \mathbb{S}^2, \tag{31}$$

where $\{\lambda_j\}_{j=0}^\infty$ are the eigenvalues of the positive definite operator $-\Delta_{\mathbb{S}^2}$, $\{\varphi_{j,k}\}_{j,k=0,1}^{\infty,2j+1}$ are the spherical harmonics (these are also eigenfunctions of $-\Delta_{\mathbb{S}^2}$), and

$$C_\nu := \sum_{j=0}^{\infty} \left( \frac{2\nu}{\ell^2} + \lambda_j \right)^{-\nu-1}, \tag{32}$$

is a normalisng constant for the Matérn spectral density on the sphere $p(j) \propto \left( \frac{2\nu}{\ell^2} + \lambda_j \right)^{-\nu-1}$. From equation 31, the weight-space view of Gaussian processes implies that $f$ is equivalent to a general linear model with deterministic feature maps of the form

$$\phi_{\mathbb{S}^2,\det}^{j,k}(s) = \sqrt{\frac{\sigma^2}{C_\nu} \left( \frac{2\nu}{\ell^2} + \lambda_j \right)^{-\nu-1}} \varphi_{j,k}(s), \quad j = 1, \dots, \infty, \quad k = 1, \dots, 2j+1. \tag{33}$$

That is,

$$f(s) = \sum_{j=0}^{\infty} \sum_{k=1}^{2j+1} \theta_{j,k} \phi_{\mathbb{S}^2,\det}^{j,k}(s) \tag{34}$$

for $\theta_{j,k} \sim \mathcal{N}(0,1)$ i.i.d. for all $j = 1, \dots, \infty$ and $k = 1, \dots, 2j+1$. This set of features can cause issues when using them in practice, since spherical harmonics have increasingly high fluctuations as $j \to \infty$, eventually not being realisable numerically due to aliasing issues. For example, using single precision floating point arithmetics, we observe that such issues occur at around the level $j = 20$. We observe that random features, which we will consider next, are better suited if we wish to keep floating arithmetics to single precision, in order to save memory.

### A.2.1 Random features on the sphere

We now derive an alternative feature map representation of Matérn GPs over $\mathbb{S}^2$ that is analogous to the random features equation 25 in the Euclidean setting. First, the addition theorem for spherical harmonics states that

$$\sum_{k=1}^{2j+1} \varphi_{j,k}(s)\varphi_{j,k}(s') = \frac{2j+1}{4\pi} \mathcal{G}_j^{1/2}\big( d_{\mathbb{S}^2}(s, s') \big), \tag{35}$$

where $d_{\mathbb{S}^2}(\cdot, \cdot)$ denotes the geodesic distance on the sphere and $\mathcal{G}_n^\alpha(\cdot)$ are the Gegenbauer polynomials. Furthermore, (Azangulov et al., 2024, Proposition 7) gives us that

$$\mathcal{G}_j^{1/2}\big(d_{\mathbb{S}^2}(s, s')\big) = \frac{2j+1}{4\pi} \int_{\mathbb{S}^2} \mathcal{G}_j^{1/2}\big(d_{\mathbb{S}^2}(s, u)\big)\mathcal{G}_j^{1/2}\big(d_{\mathbb{S}^2}(s', u)\big)\mu_{\mathbb{S}^2}(\mathrm{d}u), \tag{36}$$

where $\mu_{\mathbb{S}^2}$ is the invariant measure on the sphere. Putting all of this together gives

$$k_{\mathbb{S}^2}(s, s') = \sigma^2 \int_{\mathbb{R}} \int_{\mathbb{S}^2} \left(\frac{2\omega+1}{4\pi}\right)^2 \mathcal{G}_\omega^{1/2}\big(d_{\mathbb{S}^2}(s, u)\big)\mathcal{G}_\omega^{1/2}\big(d_{\mathbb{S}^2}(s', u)\big)\mu_{\mathbb{S}^2}(\mathrm{d}u)\mu^\nu(\mathrm{d}\omega), \quad \forall s, s' \in \mathbb{S}^2, \tag{37}$$

where $\mu^\nu(\mathrm{d}\omega)$ is the discrete measure

$$\mu^\nu(\mathrm{d}\omega) := \frac{1}{C_\nu} \sum_{j=1}^{\infty} \left(\frac{2\nu}{\ell^2} + \lambda_j\right)^{-\nu-1} \delta_j(\mathrm{d}\omega). \tag{38}$$

In practice, we consider a truncated series for *equation* 31, giving us equation 37 with the finite discrete measure

$$\mu_J^\nu(\mathrm{d}\omega) := \frac{1}{C_{\nu,J}} \sum_{j=1}^{J} \left(\frac{2\nu}{\ell^2} + \lambda_j\right)^{-\nu-1} \delta_j(\mathrm{d}\omega), \quad C_{\nu,J} = \sum_{j=0}^{J} \left(\frac{2\nu}{\ell^2} + \lambda_j\right)^{-\nu-1}, \tag{39}$$

which is equivalent to the multinomial distribution

$$\omega \sim \mathrm{Multinomial}\left(C_{\nu,J}^{-1}\left(\frac{2\nu}{\ell^2} + \lambda_1\right)^{-\nu-1}, \ldots, C_{\nu,J}^{-1}\left(\frac{2\nu}{\ell^2} + \lambda_J\right)^{-\nu-1}\right). \tag{40}$$

Thus, by Monte Carlo approximation, we have

$$k_{\mathbb{S}^2}(s, s') \approx \sigma^2 \sum_{m=1}^{M} \left(\frac{2\omega_m+1}{4\pi}\right)^2 \mathcal{G}_{\omega_m}^{1/2}\big(d_{\mathbb{S}^2}(s, u_m)\big)\mathcal{G}_{\omega_m}^{1/2}\big(d_{\mathbb{S}^2}(s', u_m)\big), \quad \forall s, s' \in \mathbb{S}^2, \tag{41}$$

with $b \in U(\mathbb{S}^2)$, the uniform distribution on the sphere, and $\omega$ is sampled according equation 40. This implies the random spherical features

$$\phi_{\mathbb{S}^2}^m(s) = \sqrt{M^{-1}c_{\omega_m}}\, \mathcal{G}_{\omega_m}^{1/2}(d_{\mathbb{S}^2}(s, b_m)), \quad s \in \mathbb{S}^2, \quad m = 1, \ldots, M, \quad c_\omega := \left(\frac{\sigma(2\omega+1)}{4\pi}\right)^2. \tag{42}$$

We may also conisder the limiting kernel $\nu \to \infty$, giving us the so-called heat kernel (Borovitskiy et al., 2020), which is analogous to the squared-exponential kernel in the Euclidean case. This is the same as before, except now the measure to sample $\omega$ reads

$$\mu_J^\infty(\mathrm{d}\omega) := \frac{1}{C_{\nu,J}} \sum_{j=1}^{J} e^{-\frac{\ell^2}{2}\lambda_j} \delta_j(\mathrm{d}\omega), \quad C_{\nu,J} = \sum_{j=0}^{J} e^{-\frac{\ell^2}{2}\lambda_j}. \tag{43}$$

From a more general perspective, on the two-sphere $\mathbb{S}^2$, we have an analogous notion of a kernel $k : \mathbb{S}^2 \times \mathbb{S}^2 \to \mathbb{R}$ to be *stationary* if it satisfies

$$k(x, y) = k(Ox, Oy), \tag{44}$$

for all $x, y \in \mathbb{S}^2$ and $O \in SO(3)$. Then a general result in (Azangulov et al., 2024) states that for *any* stationary kernel on $\mathbb{S}^2$, there exist a random feature representation, extending the classic result in Euclidean space by Rahimi & Recht (2007). In fact, this construction can be further generalised to any kernels over homogeneous spaces, which $\mathbb{S}^2$ is merely a special case of (Azangulov et al., 2024). However, for the purpose of this paper, we do not need to extend beyond the spherical setting.

# B ADDITIONAL DETAILS FOR DEEP RANDOM FEATURES

## B.1 ADDITIVE RANDOM FEATURES

In our deep neural network architecture, we add skip connections from the input layer to each bottleneck layer in order to prevent the emergence of pathological behaviours as we increase depth (Duvenaud et al., 2014). When considering spherical inputs, this requires building layers $\boldsymbol{\phi}^\ell : \mathbb{R}^B \times \mathbb{S}^2 \to \mathbb{R}^H$. We achieve this by taking $\boldsymbol{\phi}^\ell$ to be the $H$ random features of a GP $\boldsymbol{f} : \mathbb{R}^B \times \mathbb{S}^2 \to \mathbb{R}^B$ with the additive kernel $\boldsymbol{k}((\boldsymbol{x}, s), (\boldsymbol{x}', s')) = \boldsymbol{k}_{\mathbb{R}^B}(\boldsymbol{x}, \boldsymbol{x}') + \boldsymbol{k}_{\mathbb{S}^2}(s, s')$, where $\boldsymbol{k}_{\mathbb{R}^B} : \mathbb{R}^B \times \mathbb{R}^B \to \mathbb{R}^{B \times B}$ and $\boldsymbol{k}_{\mathbb{S}^2} : \mathbb{S}^2 \times \mathbb{S}^2 \to \mathbb{R}^{B \times B}$ are matrix-valued stationary kernels on the respective spaces. This corresponds to an additive GP (Duvenaud et al., 2011) $\boldsymbol{f} = \boldsymbol{f}_{\mathbb{R}^B} + \boldsymbol{f}_{\mathbb{S}^2}$, where $\boldsymbol{f}_{\mathbb{R}^B} : \mathbb{R}^B \to \mathbb{R}^B$ and $\boldsymbol{f}_{\mathbb{S}^2} : \mathbb{S}^2 \to \mathbb{R}^B$ are independent GPs with kernels $\boldsymbol{k}_{\mathbb{R}^B}$ and $\boldsymbol{k}_{\mathbb{S}^2}$, respectively. In particular, this implies layers $\boldsymbol{\phi}^\ell$ of the form

$$\boldsymbol{\phi}^\ell = \boldsymbol{\phi}^\ell_{\mathbb{R}^B} + \boldsymbol{\phi}^\ell_{\mathbb{S}^2}, \tag{45}$$

where $\boldsymbol{\phi}^\ell_{\mathbb{R}^B}, \boldsymbol{\phi}^\ell_{\mathbb{S}^2}$ are the random features corresponding to the GPs $\boldsymbol{f}_{\mathbb{R}^B}, \boldsymbol{f}_{\mathbb{S}^2}$, respectively.

Putting this together, our final deep neural network architecture $\boldsymbol{f}_{\boldsymbol{\Theta}} : \mathbb{S}^2 \to \mathbb{R}^O$ on the sphere with skip connections reads, for all $s \in \mathbb{S}^2$,

$$\boldsymbol{h}^1(s) := \boldsymbol{\Theta}^1 \boldsymbol{\phi}^1_{\mathbb{S}^2}(s), \quad \boldsymbol{h}^\ell(\boldsymbol{x}, s) := \boldsymbol{\Theta}^\ell \left( \boldsymbol{\phi}^\ell_{\mathbb{R}^B}(\boldsymbol{x}) + \boldsymbol{\phi}^\ell_{\mathbb{S}^2}(s) \right), \quad \ell = 2, \ldots, L, \quad \boldsymbol{x} \in \mathbb{R}^B, \tag{46}$$

$$\boldsymbol{f}_{\boldsymbol{\Theta}}(s) = \boldsymbol{h}^{(L)}(s), \quad \text{where} \quad \boldsymbol{h}^{(1)}(s) := \boldsymbol{h}^1(s), \quad \boldsymbol{h}^{(\ell)}(s) := \boldsymbol{h}^\ell(\boldsymbol{h}^{(\ell-1)}(s), s), \quad \ell = 2, \ldots, L, \tag{47}$$

where $\boldsymbol{\Theta}^\ell \in \mathbb{R}^{B \times H}$ for $\ell = 1, \ldots, L-1$ and $\boldsymbol{\Theta}^L \in \mathbb{R}^{B \times O}$ are trainable weights.

## B.2 FUNCTIONAL REGULARISATION

Here, we provide details on the functional regularisation that we use to regularise training of the model hyperparameters. We assume the case $O = 1$ (one output dimension) for simplicity, but this can be extended easily to multiple output dimensions. In the planar model, the regularisation term reads

$$\|\nabla f\|^2_{L^2}(t) = \int_{\mathbb{R}^I} \nabla f(\boldsymbol{x}, t) \cdot \nabla f(\boldsymbol{x}, t) \mathrm{d}\boldsymbol{x}. \tag{48}$$

The gradient is computed using PyTorch's automatic differentiation and the integral is computed using the trapezoidal rule (note: our input dimesion $I$ is typically small, e.g. two or three).

In the spherical setting, we must adapt the computations to account for the curvature of the sphere. First, we consider parameterisation of the sphere in the following spherical coordinates:

$$\text{(Longitude)} \quad \varphi \in [0, 2\pi), \tag{49}$$
$$\text{(Latitude)} \quad \theta \in [-\pi/2, \pi/2). \tag{50}$$

We consider $L^2$-inner product with respect to the Haar measure on the sphere $\mu = \cos\theta \, \mathrm{d}\theta \, \mathrm{d}\varphi$ and consider the Laplace-Beltrami operator on the sphere, which reads

$$\Delta_{\mathbb{S}^2} f = \frac{1}{\cos\theta} \frac{\partial}{\partial\theta} \left( \cos\theta \frac{\partial f}{\partial\theta} \right) + \frac{1}{\cos^2\theta} \frac{\partial^2 f}{\partial\varphi^2}. \tag{51}$$

By integration-by-parts, we get the following expression for functional regularisation on the sphere

$$\langle f, (-\Delta_{\mathbb{S}^2})f \rangle_{L^2(\mathbb{S}^2)} = -\int_0^{2\pi} \int_{-\pi/2}^{\pi/2} f(\theta, \varphi) \left( \frac{1}{\cos\theta} \frac{\partial}{\partial\theta} \left( \cos\theta \frac{\partial f}{\partial\theta} \right) + \frac{1}{\cos^2\theta} \frac{\partial^2 f}{\partial\varphi^2} \right) \cos\theta \, \mathrm{d}\theta \, \mathrm{d}\varphi \tag{52}$$

$$= \int_0^{2\pi} \int_{-\pi/2}^{\pi/2} \left( \left( \frac{\partial f}{\partial\theta} \right)^2 + \frac{1}{\cos^2\theta} \left( \frac{\partial f}{\partial\varphi} \right)^2 \right) \cos\theta \, \mathrm{d}\theta \, \mathrm{d}\varphi. \tag{53}$$

Again, the gradients are computed via automatic differentiation and the integral is computed using the trapezoidal rule.

**Remark 2** *We consider functional regularisation to regularise our hyperparameters $\boldsymbol{\lambda}$ instead of considering hyperpriors, since in deep GPs, the kernel hyperparameters in each layer are less interpretable compared to shallow GP hyperaparameters. Therefore, it becomes questionable to set a prior on the hyperparameters, when we do not really have good knowledge of them. The functional regulrisation that we use here do not require prior knowledge on the hyperparameters. Instead, it assumes prior knowledge on the corresponding model $\boldsymbol{f}_{\boldsymbol{\Theta}}(\cdot;\boldsymbol{\lambda})$, which is more realistic; for instance, we might have some prior knowledge, such as smoothness, of the ground truth field $\boldsymbol{f}^{\dagger}$ that we are trying to model. This knowledge can then be used to regularise the search for $\boldsymbol{\lambda}$, without directly imposing a prior on $\boldsymbol{\lambda}$.*

## C  EXPERIMENT DETAILS

### C.1  EVALUATION METRICS

Denote by $\mathcal{D} = \{(\boldsymbol{X}_n, \boldsymbol{y}_n)\}_{n=1}^{N}$ the training set and $\mathcal{D}^* = \{(\boldsymbol{X}_n, \boldsymbol{y}_n)\}_{n=1}^{N^*}$ be a test set, where $\boldsymbol{X}_n = (\boldsymbol{x}_n, t_n)$ denotes spatiotemporal coordinates. We consider the following metrics to evaluate our model $\boldsymbol{f}$, trained on the set $\mathcal{D}$.

**Root Mean Squared Error (RMSE).**  We first consider the root mean squared error, given by

$$\mathcal{L}_{\mathrm{RMSE}}(\boldsymbol{f}; \mathcal{D}) = \sqrt{\mathbb{E}_{(\boldsymbol{X},\boldsymbol{y})}\left[\|\boldsymbol{y} - \mathbb{E}\left[\boldsymbol{f}(\boldsymbol{X})|\mathcal{D}\right]\|^2\right]} \tag{54}$$

$$\approx \sqrt{\frac{1}{N^*}\sum_{n=1}^{N^*}\|\boldsymbol{y}_n - \mathbb{E}\left[\boldsymbol{f}(\boldsymbol{X}_n)|\mathcal{D}\right]\|^2}, \tag{55}$$

where we denote by $\mathbb{E}\left[\,\cdot\,|\mathcal{D}\right]$ the conditional expectation with respect to the event of observing the training set $\mathcal{D}$. For ensemble based models, we approximate the conditional expectation by the empirical mean of the trained models.

**Root Mean Absolute Error (RMAE).**  Similar to the RMSE score, we consider the root mean absolute error, computed as

$$\mathcal{L}_{\mathrm{RMAE}}(\boldsymbol{f}; \mathcal{D}) = \sqrt{\mathbb{E}_{(\boldsymbol{X},\boldsymbol{y})}\left[\|\boldsymbol{y} - \mathbb{E}\left[\boldsymbol{f}(\boldsymbol{X})|\mathcal{D}\right]\|\right]} \tag{56}$$

$$\approx \sqrt{\frac{1}{N^*}\sum_{n=1}^{N^*}\|\boldsymbol{y}_n - \mathbb{E}\left[\boldsymbol{f}(\boldsymbol{X}_n)|\mathcal{D}\right]\|}. \tag{57}$$

Compared to the RMSE, the RMAE is more robust to outlier values, making it a more reliable metric when the data generating model has heavier tails than Gaussian.

The RMSE and RMAE only evaluates the quality of a single statistic of the distribution $\boldsymbol{f}$ (here, we used the predictive mean). Below, we introduce metrics that also evaluates the quality of the predictive uncertainties.

**Negative Log-Likelihood (NLL).**  When we have access to the ground truth field $\boldsymbol{f}^{\dagger}$, we can use the negative log-likelihood score to evaluate how well our model's predictive distribution describes the ground truth. This is computed for each input coordinate $\boldsymbol{X}$ as

$$\mathrm{NLL}(\boldsymbol{f}(\boldsymbol{X}); \boldsymbol{f}^{\dagger}(\boldsymbol{X}), \mathcal{D}) = -\log p_{\boldsymbol{f}(\boldsymbol{X})|\mathcal{D}}(\boldsymbol{f}^{\dagger}(\boldsymbol{X})), \tag{58}$$

where $p_{\boldsymbol{f}(\boldsymbol{X})|\mathcal{D}}$ denotes the probability density function of the random variable $\boldsymbol{f}(\boldsymbol{X})|\mathcal{D}$. In particular, assuming that this is Gaussian, $p_{\boldsymbol{f}(\boldsymbol{X})|\mathcal{D}}(\boldsymbol{y}) = \mathcal{N}(\boldsymbol{y}|\boldsymbol{\mu}_{\boldsymbol{X}}, \boldsymbol{\Sigma}_{\boldsymbol{X}\boldsymbol{X}})$, we get

$$58 = \frac{1}{2}\left(\boldsymbol{f}^{\dagger}(\boldsymbol{X}) - \boldsymbol{\mu}_{\boldsymbol{X}}\right)^{\top}\boldsymbol{\Sigma}_{\boldsymbol{X}\boldsymbol{X}}^{-1}\left(\boldsymbol{f}^{\dagger}(\boldsymbol{X}) - \boldsymbol{\mu}_{\boldsymbol{X}}\right) + \frac{1}{2}\log|\boldsymbol{\Sigma}_{\boldsymbol{X}\boldsymbol{X}}| + const. \tag{59}$$

We then use the following empirical risk to assess the overall quality of uncertainty of our model $\boldsymbol{f}$

$$\mathcal{L}_{\mathrm{NLL}}(\boldsymbol{f}; \mathcal{D}) = \mathbb{E}_{\boldsymbol{X}}\left[\mathrm{NLL}(\boldsymbol{f}(\boldsymbol{X}); \boldsymbol{f}^{\dagger}(\boldsymbol{X}), \mathcal{D})\right] \tag{60}$$

$$\approx \frac{1}{N^*}\sum_{n=1}^{N^*}\mathrm{NLL}(\boldsymbol{f}(\boldsymbol{X}_n); \boldsymbol{f}^{\dagger}(\boldsymbol{X}_n), \mathcal{D}). \tag{61}$$

For ensemble based models, we use its Gaussian approximation constructed from the empirical means and variances to evaluate the NLL 59.

**Negative Log-Predictive Density (NLPD).** On the other hand, when we do not have access to the ground truth field $\boldsymbol{f}^\dagger$, we can use the negative log-predictive density to evaluate the uncertainty estimates of our model. This is given by

$$\text{NLPD}(\boldsymbol{f}; \mathcal{D}, \boldsymbol{X}, \boldsymbol{y}) = -\log \mathbb{E}_{\boldsymbol{f}|\mathcal{D}} \left[ p(\boldsymbol{y}|\boldsymbol{f}, \boldsymbol{X}) \right]. \tag{62}$$

In the case of Gaussian likelihoods, i.e., $p(\boldsymbol{y}|\boldsymbol{f}, \boldsymbol{X}) = \mathcal{N}(\boldsymbol{y}|\boldsymbol{f}(\boldsymbol{X}), \sigma^2 \boldsymbol{I})$ and Gaussian posteriors $\boldsymbol{f}|\mathcal{D} \sim \mathcal{GP}(\boldsymbol{\mu}, \boldsymbol{\Sigma})$, we have the following closed-form expression for the NLPD

$$62 = -\log \mathcal{N}(\boldsymbol{y}|\boldsymbol{\mu_X}, \boldsymbol{\Sigma_{XX}} + \sigma^2 \boldsymbol{I}) \tag{63}$$

$$= \frac{1}{2} (\boldsymbol{y} - \boldsymbol{\mu_X})^\top (\boldsymbol{\Sigma_{XX}} + \sigma^2 \boldsymbol{I})^{-1} (\boldsymbol{y} - \boldsymbol{\mu_X}) + \frac{1}{2} \log |\boldsymbol{\Sigma_{XX}} + \sigma^2 \boldsymbol{I}| + const. \tag{64}$$

Again, we use the corresponding risk to evaluate the predictive uncertainties of our model

$$\mathcal{L}_{\text{NLPD}}(\boldsymbol{f}; \mathcal{D}) = \mathbb{E}_{(\boldsymbol{X}, \boldsymbol{y})} \left[ \text{NLPD}(\boldsymbol{f}; \mathcal{D}, \boldsymbol{X}, \boldsymbol{y}) \right] \tag{65}$$

$$\approx \frac{1}{N^*} \sum_{n=1}^{N^*} \text{NLPD}(\boldsymbol{f}; \mathcal{D}, \boldsymbol{X}_n, \boldsymbol{y}_n). \tag{66}$$

As with the NLL, for ensemble based models, we use the Gaussian approximation constructed from its empirical means and variances to evaluate equation 64.

**Remark 3** *In non-conjugate models, we may consider the following upper bound of the NLPD*

$$66 \leq \mathbb{E}_{\boldsymbol{f}|\mathcal{D}} \left[ -\log p(\boldsymbol{y}^*|\boldsymbol{f}, \boldsymbol{x}^*) \right] \tag{67}$$

$$\approx -\frac{1}{J} \sum_{j=1}^{J} \log p(\boldsymbol{y}^*|\boldsymbol{f}_j, \boldsymbol{x}^*), \quad \boldsymbol{f}_j \sim p(\boldsymbol{f}|\mathcal{D}), \tag{68}$$

*where we used Jensen's inequality in the second line and Monte Carlo approximation in the last line. The resulting empirical risk reads*

$$\mathcal{L}_{\text{NLPD}}(\boldsymbol{f}; \mathcal{D}) \leq -\frac{1}{N^*} \sum_{n=1}^{N^*} \frac{1}{J} \sum_{j=1}^{J} \log p(\boldsymbol{y}_n^*|\boldsymbol{f}_j, \boldsymbol{x}_n^*), \quad \boldsymbol{f}_j \sim p(\boldsymbol{f}|\mathcal{D}). \tag{69}$$

**Continuous Ranked Probability Score (CRPS).** The continuous ranked probability score is another metric used to evaluate the quality of predicted uncertainties. Compared to the NLL or NLPD, CRPS is more robust to outliers and measures holistic calibration rather than pointwise calibration (Gneiting & Raftery, 2007).

Given a data $y \in \mathbb{R}$, the CRPS of a random variable $X \in \mathbb{R}$ for modelling the data $y$ is computed as

$$\text{CRPS}(X; y) = \int_{-\infty}^{\infty} \left( \mathbb{P}_X(X \leq x) - H(x \leq y) \right)^2 \, \mathrm{d}x, \tag{70}$$

where $\mathbb{P}_X$ denotes the law of $X$. Alternatively, one can write this as

$$\text{CRPS}(X; y) = \mathbb{E}_x \left[ |x - y| \right] - \frac{1}{2} \mathbb{E}_{x, x'} \left[ |x - x'| \right], \tag{71}$$

where $\mathbb{E}_x, \mathbb{E}_{x, x'}$ denotes expectation with respect to the laws $\mathbb{P}_X, \mathbb{P}_X \otimes \mathbb{P}_X$. This formulation can be used to evaluate the CRPS empirically using i.i.d. samples $x, x' \sim \mathbb{P}_X$. Furthermore, when $X$ is Gaussian with mean $\mu$ and standard deviation $\sigma$, we have a closed form expression of the CRPS of the form

$$\text{CRPS}(X; y) = \sigma \left( y \left( 2\Phi \left( \frac{y - \mu}{\sigma} \right) - 1 \right) + 2\phi \left( \frac{y - \mu}{\sigma} \right) - \pi^{-1/2} \right), \tag{72}$$

where $\phi$ and $\Phi$ are the probability density function and cumulative density function respectively of the standard Gaussian random variable.

Assuming that the output dimension of our model is one (i.e. $O = 1$), we get the following metric for assessing model calibration based on the CRPS

$$\mathcal{L}_{\text{CRPS}}(f; \mathcal{D}) = \mathbb{E}_{(\boldsymbol{X}, y)} \left[ \text{CRPS}(f(\boldsymbol{X}) | \mathcal{D}; y) \right] \tag{73}$$

$$\approx \frac{1}{N^*} \sum_{n=1}^{N^*} \text{CRPS}(f(\boldsymbol{X}_n) | \mathcal{D}; y_n). \tag{74}$$

## C.2 EXPERIMENT 1: EVALUATION ON A SYNTHETIC DATA

To generate our synthetic ground truth field $f^\dagger$ for this experiment, we create a MSS using 12 years of altimeter readings of sea surface height (SSH) from CS2 (2010-2022) covering the polar region densely. To de-slope the MSS we subtract the EGM2008 geoid (Skourup et al., 2017). We bin measurements on a 5x5km grid in the arctic and take the average in each bin to get an estimate of a typical polar mean SSH field. The resulting field is suitable for our assessment since it exhibits spatial non-stationarity and contains several high frequency features, which would be a challenge to recover accurately. To generate the artificial measurements, we first take nine days of space-time coordinates (between March 1st and 10th 2020) of time-evolving tracks from Sentinel-3A, 3B and CryoSat-2 satellites, then extract the value of our artificially generated mean SSH field at those spatial coordinates, and finally add i.i.d. Gaussian noise to mimic epistemic uncertainty. That is, for space-time coordinates $(\boldsymbol{x}_n, t_n)$ of the satellite track, we take

$$y_n = f^\dagger(\boldsymbol{x}_n, t_n) + \epsilon_n, \quad \epsilon_n \sim \mathcal{N}(0, \sigma_y^2), \tag{75}$$

where we set $\sigma_y = 0.01$. Our final dataset comprise 1,158,505 datapoints; we select 80% of these randomly for training and the remaining 20% for validation.

### C.2.1 TRAINING DETAILS

As usual, let $\mathcal{D} = \{(\mathbf{X}_n, y_n)\}_{n=1}^N$ denote our training data, where $\boldsymbol{X}_n = (\boldsymbol{x}_n, t_n)$ for $n = 1, \ldots, N$ are spatiotemporal coordinates of the satellite tracks. We train our model using the MSE loss, giving us the regularised empirical risk

$$\mathcal{L}_{\text{train}}(\boldsymbol{\Theta}; \mathcal{D}) = \frac{1}{N} \sum_{n=1}^N |f_{\boldsymbol{\Theta}}(\boldsymbol{x}_n, t_n) - y_n|^2 + \beta \|\boldsymbol{\Theta}\|^2, \tag{76}$$

which we minimise to find the optimal neural network weights $\boldsymbol{\Theta}$. In this experiment, we assume knowledge of the data generating process in equation 75, giving us the negative log posterior

$$-\log p(\boldsymbol{\Theta} | \mathcal{D}) = \frac{1}{2\sigma_y^2} \sum_{n=1}^N |f_{\boldsymbol{\Theta}}(\boldsymbol{x}_n, t_n) - y_n|^2 + \frac{1}{2} \|\boldsymbol{\Theta}\|^2 + const. \tag{77}$$

Comparing equation 76 with equation 77, we have the relation

$$\beta = \sigma_y^2 / N. \tag{78}$$

Assuming knowledge of the observation noise $\sigma_y$, we fix the weight decay parameter $\beta$ accordingly. We summarise the training details as follows:

- **Optimiser:** Adam
- **Learning Rate:** 0.001
- **Loss Function:** Mean Squared Error (MSE) with L2 regularisation (equation 76)
- **Number of Epochs:** 1
- **Batch Size:** 1024

### C.2.2 DRF MODEL DETAILS

For the DRF model used in this experiment, we use the following model configurations:

- **Bottleneck size:** $B = 128$ for both spatial and temporal layers.

- **Hidden unit size:** $H = 1000$ for both spatial and temporal layers.
- **Number of spatial layers:** $L_x = 1, 2, 3, 4, 10, 20$. For results in Table 1, we fix $L_x = 4$.
- **Number of temporal layers:** $L_t = 1$.

We use random features corresponding to the Matérn-3/2 kernel for both spatial and temporal layers (see Example 2 in Appendix A.1). The lengthscale and amplitude hyperparameters for both the spatial and temporal layers are tuned with Bayesian optimisation using the Python library `BoTorch` (Balandat et al., 2020) with the following configurations:

- **Spatial Lengthscale** ($\ell_{\text{spatial}}$): Optimised over the range $[0.1, 10]$
- **Temporal Lengthscale** ($\ell_{\text{temporal}}$): Optimised over the range $[0.1, 10]$
- **Amplitude** ($\sigma$): Optimized over the range $[0.1, 1]$
- **Observation Noise** ($\sigma_y$): Not optimised; set to $1 \times 10^{-2}$ in the loss function
- **Total Iterations:** 10
- **Initial Samples:** 10 (generated via Latin Hypercube Sampling)
- **Acquisition Function:** Expected Improvement (EI)
- **Surrogate Model:** Gaussian Process Regression (`SingleTaskGP`)
- **Number of Restarts for Optimisation:** 100
- **Number of Raw Samples for Initialisation:** 100

We used the same Bayesian optimisation configurations for DRF with different UQ methods (Deep Ensembles and MC Dropout).

### C.2.3 UQ DETAILS

**Deep Ensembles**   We consider 10 models to form an ensemble in this experiment. Each model in the ensemble uses the same architecture – 4 Matérn random feature layers with skip connections, each trained for 1 epoch per Bayesian optimisation iteration. The training and evaluation processes for the ensemble are parallelised using the `joblib` library, allowing efficient computation across multiple cores.

**Monte-Carlo Dropout**   We consider the same architecture (4 Matérn RFF layers) for this UQ method, however we trained it for 10 epochs to make sure the model is well-converged before dropout can be used effectively during inference. The implementation is based on the Python package `lightning-uq-box` (Lehmann, 2024).

**Variational Inference**   We consider the same architecture (4 Matérn RFF layers) for this UQ method, however we use ELBO maximisation (equation 14) for hyperparameter tuning. The implementation is based on `lightning-uq-box`.

### C.2.4 BASELINE DETAILS

**GPSat.**   `GPSat` (Gregory et al., 2024b) is a Python package for making predictions based on a mixture of local GP models, developed specifically for satellite interpolation. Individual local expert models are trained on data in subregions of the domain of interest and combined together in a post-processing step. We use expert locations spaced evenly on a 200km resolution Equal-Area Scalable Earth (EASE) grid covering the arctic region. This results in 1225 expert locations. Each local expert model uses the Sparse Gaussian Process Regression (SGPR) model in Titsias (2009) for inference, each with 500 inducing points and trained with the L-BFGS-B optimiser.

**Sparse Variational Gaussian Process (SVGP).**   We consider the sparse variational Gaussian process model in Hensman et al. (2013) with 3000 inducing points. The model is trained using the Adam optimiser with a learning rate of 0.01, a minibatch size of 1024, and 5 epochs for training. Implemented using the Python package `GPyTorch` (Gardner et al., 2018).

**Deep Gaussian Processes (DGP).** We consider a deep Gaussian process with two layers in this experiment. We use the doubly stochastic variational inference (Salimbeni & Deisenroth, 2017) for training with 512 inducing points per layer, a Monte Carlo sample size of 10 and a minibatch size of 1024. We found that increasing the number of layers did not lead to significant improvements. The model is trained using the Adam optimiser with a learning rate of 0.01 and trained for 10 epochs. Implemented using the Python package `GPyTorch`.

**ReLU MLP.** We consider a vanilla RELU MLP with 6 hidden layers, 1024 hidden units and trained with a learning rate of 0.01 for 5 epochs. We used a deep ensemble of 10 models to obtain the uncertainty estimates.

**Fourier Features Network (FFN).** Similar to SIREN, we consider a FFN with 4 layers (1 RBF RFF layer and 3 fully connected linear layers), 512 hidden units for each layer and trained with a learning rate of $1 \times 10^{-5}$ for 10 epochs. We used a deep ensemble of 10 models to obtain the uncertainty estimates. For the positional encoding, we used random Fourier features (equation 25) and employed Bayesian Optimisation to tune the kernel hyperparameters $\ell$ and $\sigma$.

**SIREN.** We consider a SIREN network with 4 hidden layers, 512 hidden units and trained with a learning rate of $1 \times 10^{-5}$ for 10 epochs. We used a deep ensemble of 10 models to obtain the uncertainty estimates.

**Conditional Neural Processes (CNP).** The encoder consists of fully connected layers with dimensions $[x_{\dim} + y_{\dim}, h_{\dim}, \ldots, r_{\dim}]$ to extract the context information, and the decoder maps this context to the target values through a series of layers $[x_{\dim} + r_{\dim}, h_{\dim}, \ldots, 2 \times y_{\dim}]$.

- **Encoder dimensions**: $[x_{\dim} + y_{\dim} = 4, 1000, 1000, 1000, 1000, 1000, 1000, 1000, 1000]$
- **Decoder dimensions**: $[x_{\dim} + r_{\dim} = 1003, 1000, 1000, 1000, 1000, 1000, 1000, 1000, 1000, 2 \times y_{\dim} = 2]$

**Remark 4** *The architectures for the neural network baselines were selected by optimising (via grid-search) on the mean NLPD score (equation 62) on the held-out validation set.*

## C.3 EXPERIMENT 2: FREEBOARD ESTIMATION FROM SATELLITE ALTIMETRY DATA

In this experiment, we use along-track data from the Sentinel-3A+B (S3A, S3B) and CryoSat-2 (CS2) satellites measuring sea-ice freeboard (in fact, the raw satellite measurement is the *radar freeboard*, which is distinct from sea-ice freeboard. This is converted to sea-ice freeboard in a postprocessing step). Freeboard measures the height of sea ice surface relative to sea level, which can be used to compute sea ice thickness (Gregory et al., 2021); the latter is important for monitoring the effect of climate change on the polar climate system.

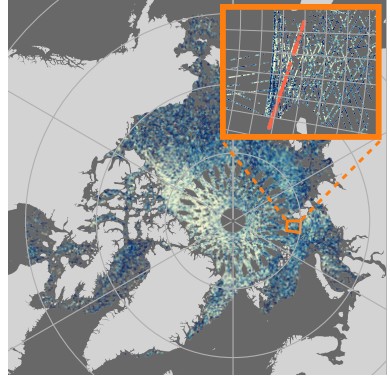 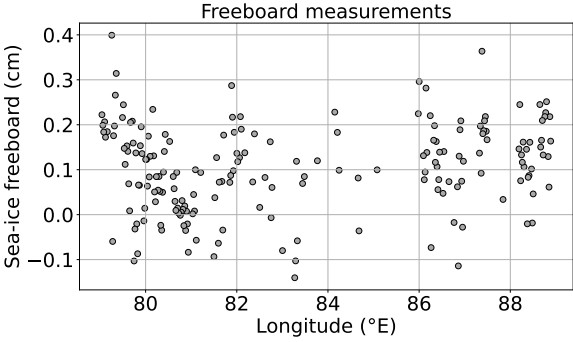

Figure 7: Freeboard measurements along a sample satellite track segment (highlighted in orange in the left figure). Real satellite measurements are typically very noisy.

We use the freeboard observations along S3A, 3B and CS2 satellite tracks downloaded from Gregory et al. (2024a). Our final dataset contains 1,174,848 data points. Since we do not have access to a ground truth field, we hold out a separate test set that we use to make our final evaluations. We use 70% of the data (randomly sampled) for training, 15% for validation and the remaining 15% for testing. The satellite measurements are extremely noisy and contain many outliers, making it a challenge to interpolate robustly, picking out the signal from the noise (see for example Figure 7 for freeboard measurements along an example track).

### C.3.1   DRF MODEL DETAILS

We use the same model architecture and Bayesian Optimisation settings as in our first experiment. For training, we also use the same procedure as in our first experiment, except we now treat the observation noise $\sigma_y$ as a tunable hyperparameter, learned jointly with the kernel hyperparameters via Bayesian optimisation.

### C.3.2   BASELINE DETAILS

**GPSat.**   We use the same configuration as in our first experiment.

**Sparse Variational Gaussian Process (SVGP).**   We use the same configuration as in our first experiment.

**Deep Gaussian Processes (DGP).**   We use the same configuration as in our first experiment.

**ReLU MLP.**   We consider a vanilla RELU MLP with 6 hidden layers, 1024 hidden units and trained with a learning rate of 0.01 for 5 epochs. We used a deep ensemble of 10 models to obtain the uncertainty estimates.

**Fourier Features Network (FFN).**   We consider a FFN with 4 layers (1 RBF RFF layer and 3 fully connected linear layers), 256 hidden units for each layer and trained with a learning rate of 0.01 for 5 epochs. We used a deep ensemble of 10 models to obtain the uncertainty estimates. For the positional encoding, we used random Fourier features (equation 25) and employed Bayesian Optimisation to tune the kernel hyperparameters $\ell$ and $\sigma$.

**SIREN.**   We consider a SIREN network with 6 hidden layers, 512 hidden units and trained with a learning rate of 0.01 for 5 epochs. We used a deep ensemble of 10 models to obtain the uncertainty estimates.

**Conditional Neural Processes (CNP).**   We use the same configuration as in our first experiment.

### C.4   EXPERIMENT 3: LARGE SCALE INTERPOLATION OF GLOBAL SEA LEVEL ANOMALY

In our final experiment, we used global sea level anomaly measurements from the Sentinel 3A satellite for the period March 1st–4th, 2020, resulting in 8,094,569 data points. We use 80% of these for training and 20% for validation.

### C.4.1   DRF MODEL DETAILS

For the DRF model used in this experiment, we use the following model configurations:

- **Number of spatial layers:** One spherical random feature layer + two Euclidean random feature layers.
- **Number of temporal layers:** $L_t = 1$.
- **Bottleneck size:** $B = 128$ for both spatial and temporal layers.
- **Hidden unit size:** $H = 1000$ for both spatial and temporal layers.

We use random features corresponding to the Matérn-3/2 kernel for both spatial (spherical and Euclidean) and temporal layers. The lengthscale and amplitude hyperparameters for both the spatial and temporal layers are tuned with Bayesian optimisation with the following configurations:

- **Spatial Lengthscale for Spherical Layer** ($\ell_{\text{spherical}}$): Optimised over the range $[1 \times 10^{-5}, 0.1]$
- **Spatial Lengthscale for Euclidean Layer** ($\ell_{\text{euclidean}}$): Optimised over the range $[1 \times 10^{-5}, 10]$
- **Amplitude for Spherical Layer** ($\sigma_{\text{spherical}}$): Optimised over the range $[1 \times 10^{-5}, 1]$
- **Temporal Lengthscale** ($\ell_{\text{temporal}}$): Optimised over the range $[1 \times 10^{-5}, 10]$
- **Amplitude for Euclidean Layer** ($\sigma_{\text{euclidean}}$): Optimised over the range $[1 \times 10^{-5}, 1]$
- **Total Iterations:** 10
- **Initial Samples:** 15 (generated via Latin Hypercube Sampling)
- **Acquisition Function:** Expected Improvement (EI)
- **Surrogate Model:** Gaussian Process Regression (`SingleTaskGP`)
- **Number of Restarts for Optimisation:** 100
- **Number of Raw Samples for Initialisation:** 100

We train the model with a Huber loss

$$\ell_{\text{Huber}}(\boldsymbol{f}; \boldsymbol{X}, \boldsymbol{y}, \delta) = \left\{ \begin{array}{l} \frac{1}{2}|\boldsymbol{f}(\boldsymbol{X}) - \boldsymbol{y}|^2 \\ \delta(|\boldsymbol{f}(\boldsymbol{X}) - \boldsymbol{y}| - \delta/2) \end{array} \right. \tag{79}$$

with $\delta = 0.1$. This loss is more robust than the mean squared error to outlier values.

### C.4.2 BASELINE DETAILS

**Spherical SVGP.** We used the spherical Matérn-3/2 GP (Borovitskiy et al., 2020) with 3000 inducing points, trained using the Adam optimiser with a learning rate of 0.01 and a minibatch size of 1024. Implemented using the Python package `GPyTorch` with the spherical kernel implemented with the `geometric-kernels` package.

**Euclidean DRF.** We used a Euclidean DRF model with two layers, each with bottleneck dimension $B = 128$ and hidden dimension $H = 1000$. We also utilised Bayesian Optimisation to tune the kernel hyperparameters $\ell_{\text{spatial}}$, $\ell_{\text{temporal}}$ and $\sigma$.

### C.4.3 QUANTITATIVE RESULTS

| Model | NLPD | CRPS |
|---|---|---|
| Spherical DRF | **$0.0063 \pm 0.0002$** | **$0.0918 \pm 0.0021$** |
| Euclidean DRF | $0.0387 \pm 0.0001$ | $0.1158 \pm 0.0008$ |
| SVGP | $0.0085 \pm 0.0001$ | $0.1210 \pm 0.0010$ |

Table 3: NLPD and CRPS comparison for Spherical DRF, Euclidean DRF and SVGP Models

In Table 3, we present the quantitative results for experiment 3, comparing the performance of spherical DRF, Euclidean DRF and spherical SVGP models using the NLPD and CRPS metrics. For the NLPD computation, we use its approximation (equation 69) with the likelihood defined by the Huber loss. That is, $p(\boldsymbol{y}|\boldsymbol{f}, \boldsymbol{X}) \propto \exp\left(-\ell_{\text{Huber}}(\boldsymbol{f}; \boldsymbol{X}, \boldsymbol{y}, \delta)\right)$.

Consistent with what we see in the qualitative performance (Figure 6), the spherical DRF significantly outperforms the other baselines, achieving the lowest NLPD and CRPS.

## C.5   GALLERY OF PREDICTIONS

### C.5.1   GALLERY OF PREDICTIONS: EXPERIMENT 1

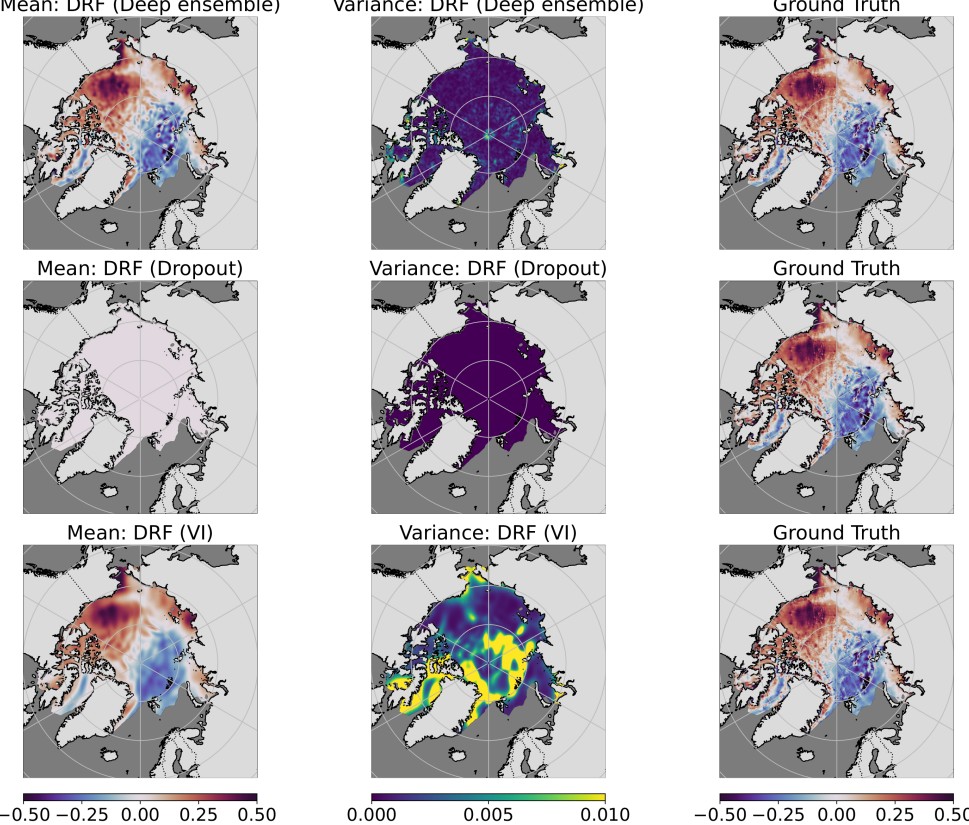

Figure 8: Comparison of predictive means and variances of the DRF model using various UQ methods on the synthetic experiment. Left column: predictive means, Middle column: predictive variances, Right column: ground truth MSS field.

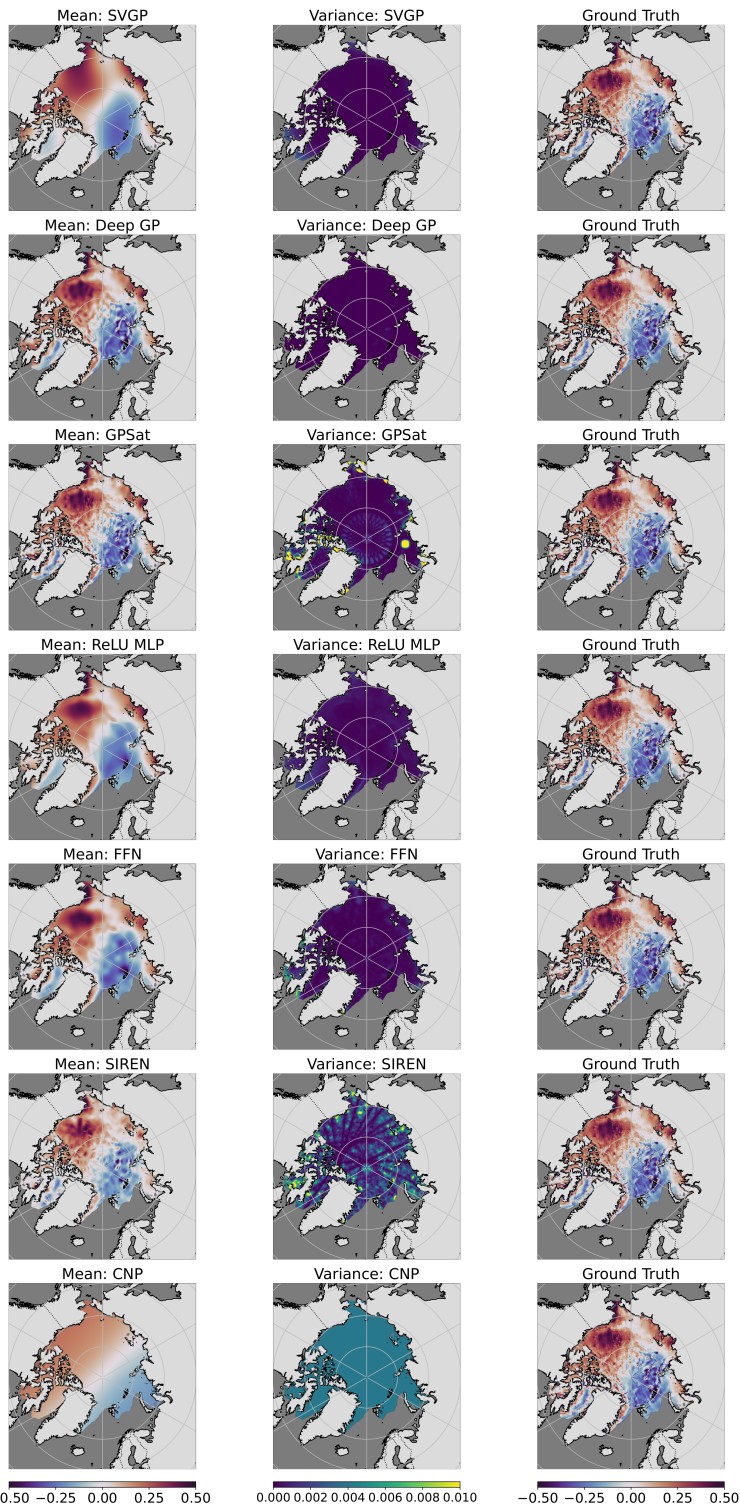

Figure 9: Comparison of the baseline models on the synthetic experiment. Left column: predictive means, Middle column: predictive variances, Right column: ground truth MSS field.

### C.5.2 GALLERY OF PREDICTIONS: EXPERIMENT 2

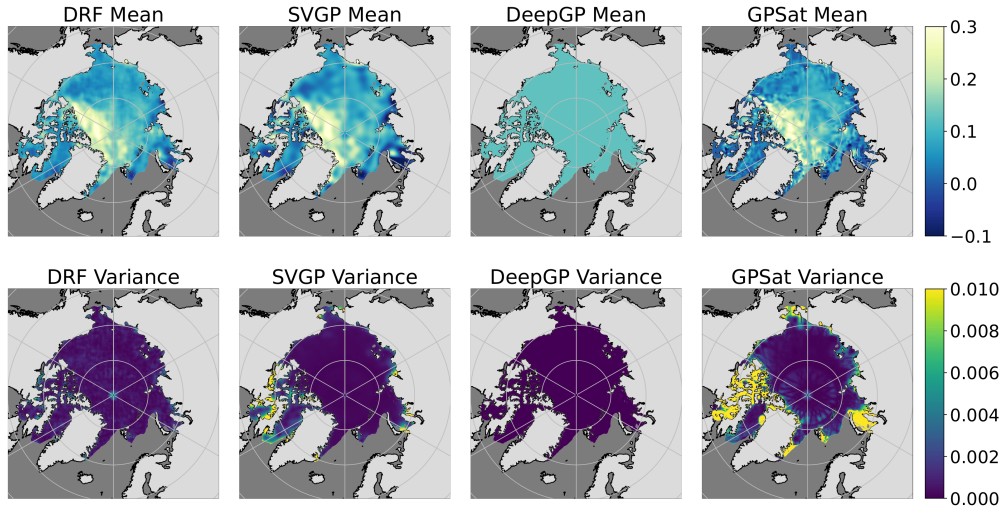

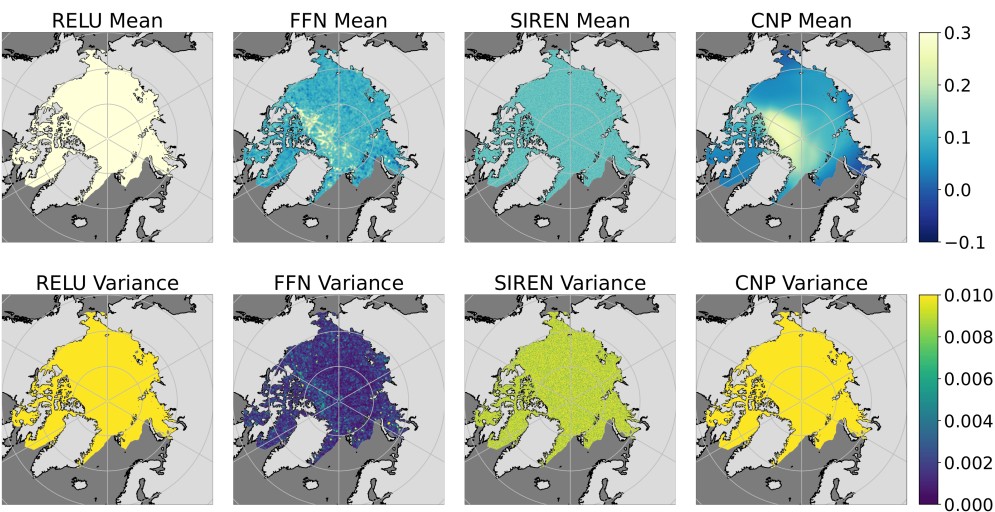

Figure 10: Comparison of DRF and other baseline models on freeboard interpolation. We display both predictive means and variances for each model.

### C.5.3 GALLERY OF PREDICTIONS: EXPERIMENT 3

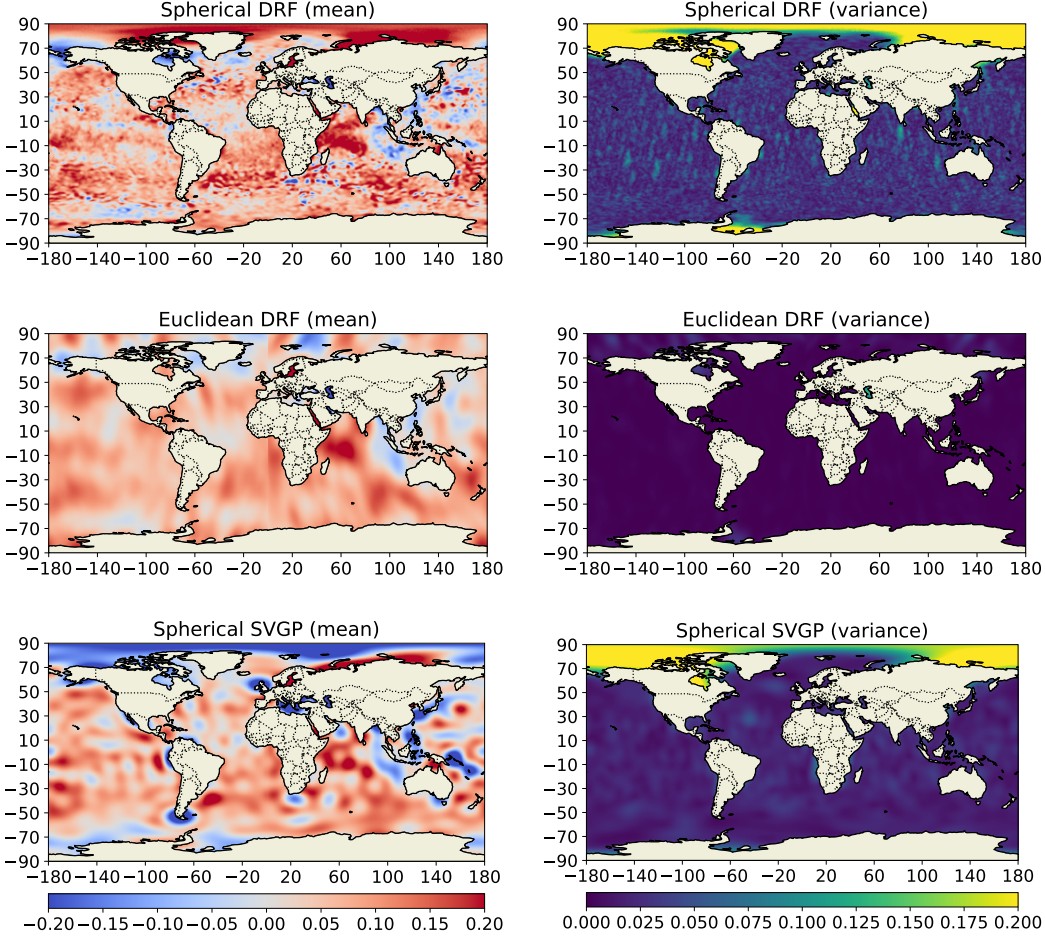

Figure 11: Result maps (mean and variance) for global sea level anomaly interpolation. From top to bottom: Spherical DRF, Euclidean DRF and Spherical SVGP.

## C.6 FURTHER ABLATION STUDY

### C.6.1 DATA SCALABILITY

Here, we conducted an experiment where we trained on 10% and 1% subsampled data of our global sea level anomaly interpolation experiment. The dataset partition follows the following ratios: 70% training, 15% validation and 15% testing splits to compute the metrics. The results can be found in Table 4 below where we compare our DRF model against SVGP (3,000 inducing points, trained for 2 epochs).

This show that, as expected, the DRF model achieves better performance with increasing data size, as evidenced by the lower NLPD and CRPS values. We also find that for larger data (10% and 100%), DRF performs better than SVGP due to its ability to capture finer and finer scale features as we increase data resolution (note that the performance of SVGP does not change much by increasing the data from 10% to 100%). On the other hand, for smaller data proportions (1%), the simpler inductive bias in the SVGP helps to generalise better, resulting in better performance compared to DRF. See figure 12 for comparisons.

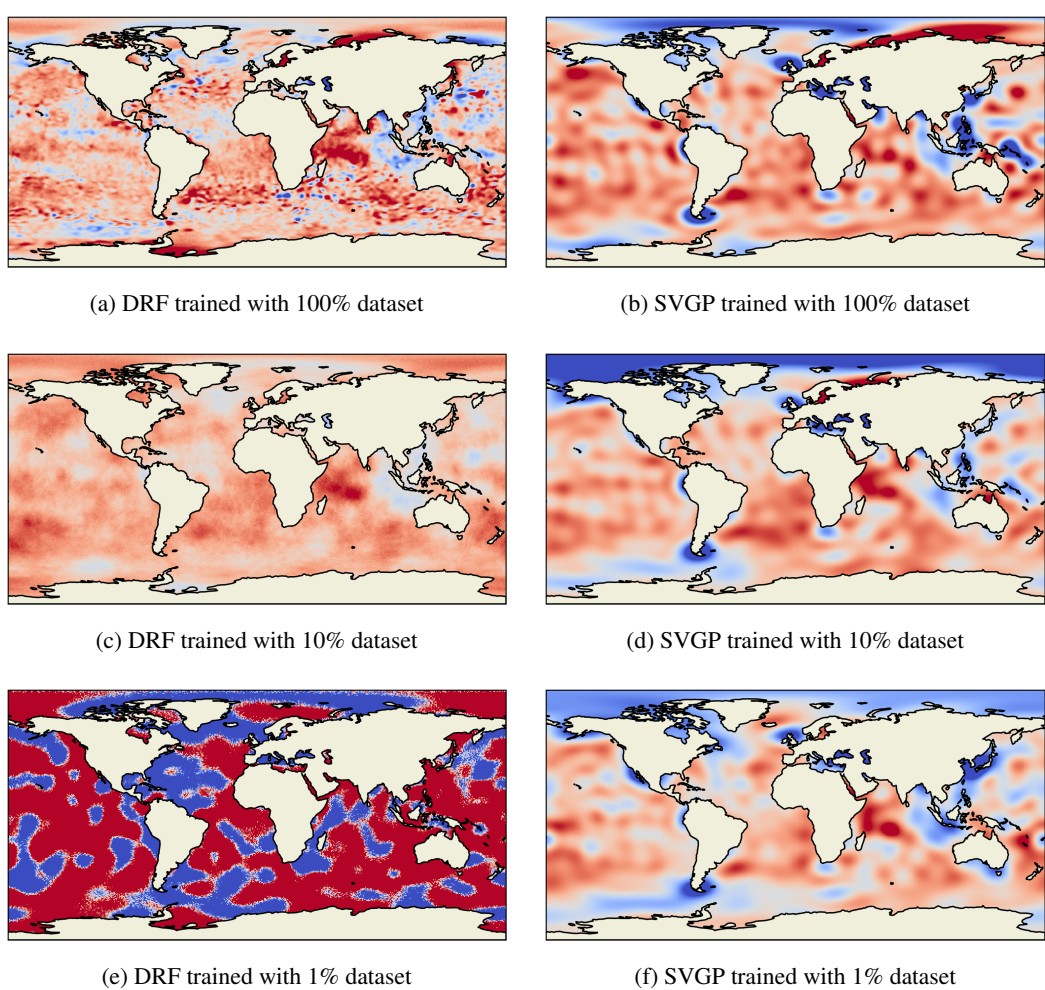

(a) DRF trained with 100% dataset

(b) SVGP trained with 100% dataset

(c) DRF trained with 10% dataset

(d) SVGP trained with 10% dataset

(e) DRF trained with 1% dataset

(f) SVGP trained with 1% dataset

Figure 12: Results maps shows data scalability and corresponding performances for DRF (left column) and SVGP (right column)

### C.6.2 SKIP CONNECTIONS

Below, we show how the addition of the skip connections affect our global sea level anomaly interpolation results. We find that without the skip connections, we get unstable results, sometimes

| Dataset | NLPD | CRPS | Time (mins) |
|---|---|---|---|
| Full Dataset-DRF | **0.0063** | **0.0927** | 291.15 |
| Subsampled-DRF (1 in 10) | 0.0079 | 0.1114 | 45.61 |
| Subsampled-DRF (1 in 100) | 0.1314 | 0.6490 | 17.82 |
| Full Dataset-SVGP | 0.0085 | 0.1205 | 345.04 |
| Subsampled-SVGP (1 in 10) | 0.0086 | 0.1200 | 47.46 |
| Subsampled-SVGP (1 in 100) | 0.0161 | 0.1435 | 18.10 |

Table 4: Comparison of NLPD, CRPS, and Time for Various Dataset Sizes for DRF and SVGP

leading to poor reconstructions of the field as illustrated in the left figure 13a. We display the corresponding metrics in Table 5, where we see that the model with the skip connection produce better scores as a result.

For our other experiments, we did not see major differences between the model with or without skip connections.

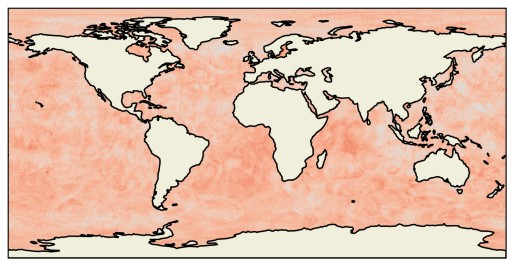 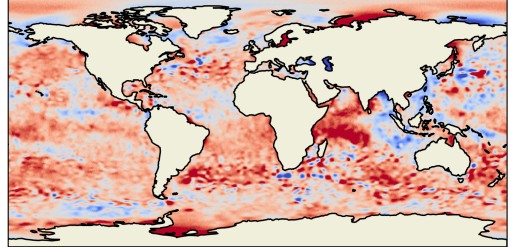

(a) Example of poor result when using DRF model without skip connections

(b) Typical prediction of DRF model with skip connections

Figure 13: Performances of DRF models with and without skip connections

| Model | NLPD | CRPS |
|---|---|---|
| Model with skip connections | **0.0063 ± 0.0002** | **0.0918 ± 0.0021** |
| Model without skip connections | 0.0078 ± 0.0007 | 0.1092 ± 0.0101 |

Table 5: Comparison of NLPD, CRPS, for model with or without skip connections

### C.6.3 SCALABILITY WITH MODEL WIDTH

Here, we present the results of ablation experiment with respect to model width defined as the number of random feature per layer. In the first part, we fixed the bottleneck size $B$ per layer and varied the number of hidden units $H$ between the bottlenecks, while keeping the model depth fixed at 3. The number of hidden units $H$ directly corresponds to the number of random features per layer. Our findings indicate that, in general, increasing the number of random features (i.e., enlarging $H$) leads to improved performance, as shown in Table 6.

| DRF Model | NLPD | CRPS |
|---|---|---|
| $H = 1000, B = 128$ | **0.0063 ± 0.0002** | **0.0918 ± 0.0021** |
| $H = 500, B = 128$ | 0.0064 ± 0.0001 | 0.0944 ± 0.0009 |
| $H = 250, B = 128$ | 0.0065 ± 0.0001 | 0.0951 ± 0.0006 |

Table 6: NLPD and CRPS Comparison for DRF models with different hidden size

In the second part, we have conduced an ablation experiment with respect to the bottleneck size $B$ while hidden size held fixed ($H = 1000$).

| DRF Model | NLPD | CRPS |
|---|---|---|
| $H = 1000, B = 32$ | 0.0066 ± 0.0001 | 0.0969 ± 0.0006 |
| $H = 1000, B = 64$ | 0.0067 ± 0.0003 | 0.0969 ± 0.0030 |
| $H = 1000, B = 128$ | **0.0063 ± 0.0002** | **0.0918 ± 0.0021** |
| $H = 1000, B = 1000$ | 0.0065 ± 0.0003 | 0.0932 ± 0.0035 |

Table 7: NLPD and CRPS Comparison for DRF models with different bottleneck size

We also observe that increasing the bottleneck size generally improves performance, but this only holds true up to a certain point. Beyond that, such as when we set $B = 1000$, the results actually begin to deteriorate, and the hyperparameter tuning process becomes less stable. Figure 14 shows an example of poor results we got when we set $H = 1000, B = 1000$.

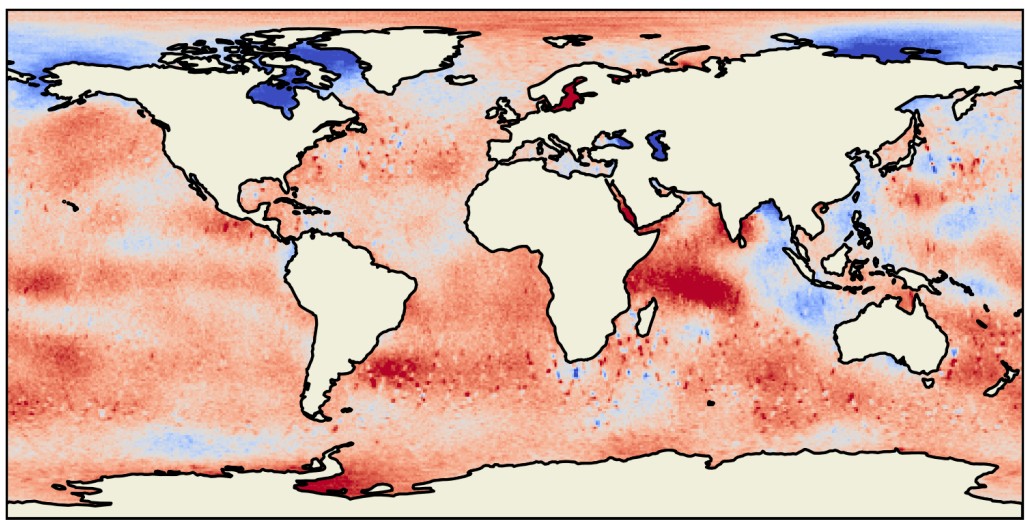

Figure 14: Example of poor result of DRF model with $H = 1000, B = 1000$

