# OpenReview forum: "Deep Random Features for Scalable Interpolation of Spatiotemporal Data"
_ICLR.cc/2025/Conference — ICLR 2025 Poster_

### Official Review · Reviewer_jnCy · 2024-10-22

**Soundness:** 4
**Presentation:** 4
**Contribution:** 3
**Rating:** 8
**Confidence:** 4

**Summary:**

The paper proposes to blend the strengths of neural networks (NNs) and Gaussian processes (GPs) by introducing the deep random features (DRF) framework, where the weights of the Fourier expansion $\boldsymbol{\Theta}$ are viewed as trainable parameters (similar to weights in NN) and the random features themselves are viewed as neurons. To perform interpolation on sphere, the authors also construct random feature maps based on a kernel constructed by a Mercer sum. The hyperparameters of the model are tuned either by comparing the evidence lower bound (ELBO) or by using a held-out validation set, depending on the choice of uncertainty quantification method. The authors test DRF on both synthetic and remote sensing datasets to show that DRF is able to learn both coarse-scale and fine-scale data patterns with well-calibrated uncertainty.

**Strengths:**

•	The authors present the methodology clearly, supplemented by informative visualizations, and provide a comprehensive discussion of the experimental results.

•	The proposed framework integrates the flexibility of NNs with the inductive bias of GPs, offering a scalable solution for large datasets.

•	Detecting high-frequency patterns is important in environmental data analysis, as such patterns are often closely associated with extreme events.

•	While there exists some amount of literature on constructing random features from kernel machines, the application of random features for interpolating spatial and temporal data is relatively underexplored and can be considered novel.

**Weaknesses:**

•	For clarity, it would be better to differentiate $H$ from $B$ in Section 3.1, as $H$ plays a similar role as $M$ in Section 2.2 which represents the dimension of the random features, while the bottleneck dimension $B$ is analogous to the hidden layer dimension in NNs.

•	A more comprehensive discussion of related work would enhance the paper’s contribution. For instance, [1] investigated representation learning through the approximation of kernels using random Fourier features. Additionally, [2] explored the connection between deep ensembles trained with squared-error loss and GP posteriors.

•	As noted in Section 2.2, random Fourier features are applicable only to stationary kernel functions. However, spatiotemporal data may exhibit non-stationary trends, which presents a potential limitation of the proposed method. A discussion on this point (e.g., extensions to handle non-stationarity) will be appreciated.

References:

[1]. Jiang, Z., Zheng, T., Liu, Y., & Carlson, D. (2022). Incorporating prior knowledge into neural networks through an implicit composite kernel. arXiv preprint arXiv:2205.07384.

[2]. He, B., Lakshminarayanan, B., & Teh, Y. W. (2020). Bayesian deep ensembles via the neural tangent kernel. Advances in neural information processing systems, 33, 1010-1022.

**Questions:**

•	Since $\boldsymbol{\omega} \sim p(\boldsymbol{\omega})$ and $p(\boldsymbol{\omega})$ is the normalized Fourier transform of the kernel function, can you elaborate the procedure to efficiently sample from $p(\boldsymbol{\omega})$?

•	Why does DRF with dropout have much higher computational time than an ensemble of DRFs?

•	Does it really make sense to compare with neural processes in Tables 1 and 2 as NPs are mainly used for meta-learning?

•	Are ground truth values available for the sea level anomaly measurements? If so, it would be beneficial to present the interpolation and uncertainty calibration errors to validate the importance of learning fine-scale fluctuations.

---

> ### Author Response · Authors · 2024-11-16
> **Addressing the weaknesses**
>
> __We thank the reviewer for going through our work and providing useful comments. We are glad that you found our presentation to be clear and the visualisations to be informative. Please find our individual responses below to the various points raised regarding weaknesses:__
>
> > For clarity, it would be better to differentiate $H$ from $B$ in Section 3.1, as $H$ plays a similar role as $M$ in Section 2.2
>
> Thank you for pointing this out. We agree that the notations are inconsistent -- we've used $M$ and $H$ interchangeably. We will stick to just using $H$, which emphasises that this corresponds to the size of the hidden layer.
>
> > A more comprehensive discussion of related work would enhance the paper's contribution...
>
> Thank you for making us aware of these works. We will add the references in appropriate places of the manuscript to place the work in better context.
>
> > As noted in Section 2.2, random Fourier features are applicable only to stationary kernel functions...
>
> Please see our response to Reviewer pDLc for details regarding this. In short, our model is actually non-stationary since we are composing stationary layers, and the composition of stationary features are not stationary.

---

> > ### Author Response · Authors · 2024-11-16
> > **Addressing the questions**
> >
> > > Since $\boldsymbol{\omega} \sim p(\boldsymbol{\omega})$ and $p(\boldsymbol{\omega})$ is the normalized Fourier transform of the kernel function, can you elaborate the procedure to efficiently sample from $p(\boldsymbol{\omega})$?
> >
> > In the cases that we have considered (using kernels of the Mat\'ern family), $p(\omega)$ is just the normal or student $t$ distribution so it is very easy to sample from (see Appendix A.1 for details). In the spherical case, the distribution corresponding to $p(\omega)$ becomes a multinomial distribution with weights determined by the kernel (which we know in closed form, again for the Mat\'ern family). Again, this is easy to sample from and deep learning libraries such as PyTorch already have this implemented.
> >
> > > Why does DRF with dropout have much higher computational time than an ensemble of DRFs?
> >
> > The reason why the dropout model takes more time is that, by nature of the dropout procedure, it simply requires a larger number of epochs to train in the inner optimisation loop. This is due to the fact that the model has to encounter various dropout configurations during the training phase to learn the weights. In comparison, without dropout, we find that usually one training epoch is sufficient to achieve convergence, due to the small network size. Details of the training are provided in Appendix C.2.3.
> >
> > > Does it really make sense to compare with neural processes in Tables 1 and 2 as NPs are mainly used for meta-learning?
> >
> > Indeed, neural processes are used for meta-learning. However, this is distinct from conditional neural processes, which is what we use as a baseline. Conditional neural processes are primarily used for similar tasks (interpolation with UQ) as what we consider in this work.
> >
> > > Are ground truth values available for the sea level anomaly measurements? If so, it would be beneficial to present the interpolation and uncertainty calibration errors to validate the importance of learning fine-scale fluctuations.
> >
> > Unfortunately, no ground truth values are available for our last two experiments. The best we can do is to compute results on held out observations, which are typically very noisy, hence not entirely reliable. Therefore, in our submission, we had only performed qualitative assessment of our third experiment, which we felt was sufficient. However, we have now performed quantitative assessment, similar to our second experiment, where we computed the NLPD and CRPS on randomly held out observations. The results are given in the table below, which does in fact demonstrate that the DRF model produce better results compared to SVGP, suggesting that the fine-scale fluctuations we see are true signals.
> >
> > | Model   | NLPD | CRPS |
> > |----------|----------|----------|
> > | DRF | __0.0063 ± 0.0002__ | __0.0918 ± 0.0021__ |
> > | SVGP | 0.0085 ± 0.0001 | 0.1210 ± 0.0010 |
> >
> > In the future, we are considering cross comparing with observations coming from different instruments to further validate our claim.

---

> > > ### Comment · Reviewer_jnCy · 2024-11-20
> > >
> > > Thank you for your reply to my comments and questions. I think the authors' response have addressed most of my concerns, and I will keep my score for now.

---

> > > > ### Author Response · Authors · 2024-11-22
> > > > **Thank you**
> > > >
> > > > Thank you for your time and effort to review our work. We will make sure to clarify all of the above points in our updated manuscript.

---

### Official Review · Reviewer_pDLc · 2024-10-27

**Soundness:** 2
**Presentation:** 2
**Contribution:** 2
**Rating:** 3
**Confidence:** 4

**Summary:**

The paper presents a deep Gaussian process (GP) with deep random features for modeling spatiotemporal data where the inputs are spatial/temporal coordinates. Deep random features are utilized to address the high-frequency limitations of neural networks, and mini-batched gradient descent is used for large-scale training. Experiments are conducted on the synthetic data, satellite data, and sea level anomaly data.

**Strengths:**

- The paper addresses a clear and timely topic, focusing on coordinate-based neural representation for sparse observations.
- Figures 1 and 2 effectively illustrate both the task and the model structure.
- The theoretical background on random features is covered in sufficient depth.

**Weaknesses:**

- The model does not appear to outperform baselines in terms of accuracy or computation time.
- Although the primary contribution is the application of random features, the paper does not cover advanced Fourier, Wavelet features, such as [1]. It would be helpful for the authors to consider incorporating these features or at least to discuss how their approach might be extended to include them.
- The assumption of stationary kernel seems limit model performance, and further clarification on this assumption would be beneficial.
- It would be beneficial for the authors to include tasks for both time interpolation and extrapolation, like [2].
- The paper also overlooks many recent works that apply implicit neural representations for spatiotemporal data, which should be included for a more comprehensive comparison. For the existing works, temporal dimensions are often treated separately from spatial dimensions, typically using autoregressive time-stepping, which has shown efficiency in modeling physical spatiotemporal data. For example, [2], and [3] treat the spatial dimension continuously and employ neural ODEs to model time as a continuous variable. The author should justify why the proposed method has advantages over existing continuous spatial-temporal models.

[1] Multiplicative Filter Networks, ICLR, 2021.

[2] Continuous PDE Dynamics Forecasting with Implicit Neural Representations, ICLR, 2023.

[3] Operator Learning with Neural Fields: Tackling PDEs on General Geometrics, Neurips, 2023.

**Questions:**

See above.

**Details Of Ethics Concerns:**

See above.

---

> ### Author Response · Authors · 2024-11-16
> **Addressing the weaknesses and clarifications about the paper**
>
> __We appreciate the reviewer for the feedbacks. Please find below our targeted responses to the various concerns raised. We hope that this will clarify all the points made by the reviewer about our results and contributions.__
>
> > The model does not appear to outperform baselines in terms of accuracy or computation time.
>
> We would like to reiterate the main points of our results as, we admit, is quite subtle and our explanation in the paper may have been insufficient.
>
> - In the first experiment, we evaluated our model on a synthetic data, where we have access to the ground truth and control over observation noise. The point here is that we are able to see how accurately the models can reconstruct the ground truth (see Figure 8 in supplmentary materials).
> Now, the only real competitors we find here in terms of performance (both qualitatively and quantitatively) are the DGP and GPSat baselines. However, both of these approaches come with more computational cost than our DRF ensembles model (in fact, if we omit functional regularisation, which gives us similar results in this case, the computation cost of DRF is further reduced, usually to under 10 minutes). The cheaper baselines all perform worse than DRF qualitatively and also in some metrics, in particular, CRPS and RMSE. We note the unreliability of the NLL metric, which seems to give preference to underconfident models. For example, DRF (VI) and CNP give the top two NLL performances, however, their qualitative results are sub par (see Figure 8 in supplementary). Altogether, the advantage of our model in this setting should be quite clear.
>
> - Regarding the second experiment, we evaluate the models on a real dataset. In this setting, we don't have access to a ground truth so we can only rely on quantitative results, which is also not completely trustworthy due to the amount of noise corrupting the data.
> That said, if we believe in the quantitative results, we find the baselines that are competitive are the SVGP and GPSat models. GPSat is again expensive (same cost as the first experiment), and upon closer inspection of the qualitative result, produces some unnatural artifacts in their predictions. SVGP on the other hand appears to perform well with similar cost as DRF on this experiment, due to the dominance of lower frequency features in the underlying field -- a setting where SVGP shines. Thus, DRF in this setting is competitive to SVGP; but note that SVGP does not perform very well on our other experiments. Thus, aside from the expensive GPSat model, DRF is the only model that consistently produced good results across all experiments (we have now also computed metrics for the third experiment, which further demonstrates this case. See our response to reviewer VtAL).
>
> If the reviewer has any suggestions to help clarify these points in our text, we would be happy to incorporate those.
>
> > Although the primary contribution is the application of random features, the paper does not cover advanced Fourier, Wavelet feature...
>
> With all due respect, we do not really see good reason to consider advanced features for the sake of incorporating these. The main purpose of our work is to develop scalable models for spatiotemporal interpolation with calibrated uncertainties. For this purpose, we take inspiration from GPs / deep GPs regression, which are well-suited for this purpose (although the suitability of deep GPs can be debatable) and combining it with recent ideas in neural representation. The Fourier features that we use (and the Gegenbauer polynomial features for the spherical case) are simply a result of them being derived from the random feature expansion of GP layers. Hence, we would expect that in the limit $H \rightarrow \infty$, each layer in our model has the same inductive bias as the GP; our hypothesis was that the model can benefit from the generalisation capability of GPs, which we would not get by considering arbitrary features.
>
> > The assumption of stationary kernel seems limit model performance, and further clarification on this assumption would be beneficial.
>
> We would like to point out that our model is in fact non-stationary since the composition of stationary features are no longer stationary. One can even see this visually (see our results gallery in the Appendix), where the DRF model is able to fit patterns across different spatial scales. The per-layer stationarity is imposed primarily for (1) its ability to be approximated by random features, and (2) its generalisation capability. Hence, we do not see this as a limitation but rather as adding benefit to our model architecture. We will make this clearer in the text.

---

> > ### Author Response · Authors · 2024-11-16
> > **Continuation of response**
> >
> > > It would be beneficial for the authors to include tasks for both time interpolation and extrapolation, like [2].
> >
> > We should point out that we actually do consider time interpolation - the held out validation set that we use to compute our results come from satellite tracks at different times so we do in fact check our model's capability to generalise across space and time. Regarding extrapolation, this is beyond the scope of what our model is capable of performing, since the inductive bias of deep GPs will not help to generalise outside of the training data distribution. To achieve the latter, it will become necessary to learn the temporal evolution operator, requiring full-field observations at each time step, which we normally do not have in remote sensing.
> >
> > > The paper also overlooks many recent works that apply implicit neural representations for spatiotemporal data, which should be included for a more comprehensive comparison...
> >
> > We thank the reviewer for pointing out this line of works. However, we do not believe the frameworks in [2], [3] are really appropriate for our setting, due to differences in data assumptions, as mentioned earlier. For example, in both works [2] and [3], the observations at each snapshot of time is assumed to lie on a space-covering grid (albeit irregular), which is sensible for them as their goal is to learn mapping between fields, which can be rolled-out autoregressively to produce spatiotemporal predictions. For the setting we are considering however, we only have raw observations from satellite tracks, where at each time snapshot, we only have a slither of observations, which barely cover the globe (this may have been misleading as Figure 1 in our paper shows observations accumulated over 4 days). It is hard to believe that the methods in [2] and [3] are able to learn anything from such small observations per time step. Our model handles this via the built-in inductive biases coming from the GP layers, much like in traditional kriging.

---

> ### Author Response · Authors · 2024-11-22
> **Follow-up on rebuttal responses**
>
> We sincerely appreciate the time and effort you've dedicated to reviewing our manuscript. As the discussion phase is nearing its conclusion, we would like to kindly confirm whether our responses have addressed and resolved any questions or concerns you may have.
> If you require any further clarification or have additional points you'd like to discuss, please do not hesitate to reach out. We are more than happy to continue the dialogue and ensure all aspects are thoroughly addressed.
> Thank you again for your valuable input.

---

> ### Author Response · Authors · 2024-11-29
> **Re: Follow up on the rebuttal response**
>
> We wanted to follow up again regarding our responses to your comments since we haven't heard from you. We only have a few days left for the discussion and we want to ensure that our replies have addressed your concerns and questions adequately.
> If the reviewer believes our responses are not satisfactory, please let us know so that we can help clarify further in the remaining discussion period.

---

### Official Review · Reviewer_VtAL · 2024-10-27

**Soundness:** 3
**Presentation:** 4
**Contribution:** 3
**Rating:** 8
**Confidence:** 4

**Summary:**

The paper introduces a model leveraging deep random features (DRF) to achieve scalable interpolation for large-scale, high-dimensional spatiotemporal data. By integrating random Fourier feature layers with DNNs, the model retains the inductive biases of GPs, while with potential for scaling efficiently with the dataset size. The novelty lies primarily in adapting random feature layers to produce an architecture capable of high-resolution spatiotemporal interpolation with uncertainty quantification, moving beyond the limitations of traditional GPs and recent DGPs in this domain.

**Strengths:**

- The paper integrates multiple uncertainty quantification techniques (variational inference, dropout, deep ensembles), offering flexibility in obtaining uncertainty estimates for the model outputs.
- The architecture performs well across varied datasets, including synthetic, local, and global satellite measurements, and the spherical adaptation for global data fields demonstrates its versatility.

**Weaknesses:**

I don’t see any major concerns or weaknesses in this paper, but I believe it would benefit from some ablation studies, which are currently missing, to better illustrate the contributions of the proposed model. Please refer to the questions for details.

**Questions:**

- To me, the proposed method closely resembles the Fourier features network and SIREN in various aspects. Specifically, it feels like a generalization of Tancik et al.’s repeated Gaussian mapping followed by linear transformations, applied across layers rather than in just the input. If random features were only used in the first layer, would the network’s performance still mirror that of the Fourier features network? Exploring this could clarify if deeper layers with random features add unique value.
- The model also feels akin to a generalized SIREN, with deterministic parameter layers replaced by additional linear transformations and stochastic components. In this context, I'd appreciate a breakdown of the number of trainable parameters in this model versus SIREN and other baselines. This would help me interpret differences in the quantitative results presented in Tables 1 and 2.
- Since the model’s roots are claimed to be in deep Gaussian processes, I think it would be helpful to see an ablation study comparing deep GPs approximated with random features against this proposed model. This doesn’t need to be overly complex—a simple synthetic dataset might be sufficient to highlight the supposed benefits of deep random features combined with learnable linear layers.
- I find the skip connections interesting but somewhat confusing. From what I see, after the projection by $\phi$, the features are concatenated back to a space seemingly compatible with the original coordinates. However, I’m unsure if this compatibility holds without explicit regularization. An ablation study on removing or adjusting these skip connections would help clarify their impact, making it easier to interpret how well they function as intended.
- The paper highlights scalability as a major benefit, but it lacks empirical validation in terms of how the model performs with increasing data points or the number of random features. Scalability results as a function of data volume and feature count would substantiate claims about the model’s efficiency and approximation performance.
- Lastly, given the reliance on random sampling for the random Fourier features, I want to see the variability in the model’s performance. Not on a UQ sense but evaluating the model across multiple runs with varied random seeds.

---

> ### Author Response · Authors · 2024-11-16
> **Addressing the questions**
>
> __Thank you for taking the time for going through our manuscript and providing useful suggestions that will help strengthen our work. Please find our responses below for each of your questions:__
>
> > To me, the proposed method closely resembles the Fourier features network and SIREN in various aspects...
>
> Yes, as the reviewer points out, our model is closely related to the use of Fourier features in SIREN and other similar architectures, which we took as initial inspirations. For Tancik et al., are you referring to the following work?
>
> Tancik et al. "Fourier Features Let Networks Learn
> High Frequency Functions in Low Dimensional Domains", NeurIPS 2020
>
> Here, they apply Fourier-based positional encoding of the input coordinates before passing it through a standard ReLU MLP. If this is what the reviewer is suggesting, we are happy to compare our model against it.
>
>
> > The model also feels akin to a generalized SIREN...
>
> Indeed, our model can be understood as an extension of SIREN with a few distinctions: (1) The architecture is slightly different, where we have a linear layer following the Fourier features, leading to a bottleneck structure; (2) We only train the linear layers with the parameters of the Fourier layers held fixed. Since we are only training the linear layers, the parameter count of our DRF model is usually smaller than a similar width and depth MLP model. For example, in our first experiment, we have the following parameter counts:
>
> - DRF model (4 layers, 1000 hidden dim, 128 bottleneck dim, 1 output dim): Total number of trainable parameters is 640,897
>
> - SIREN 4 layers. 512 hidden dim, 1 output dim): Total number of trainable parameters is 1,578,497
>
> - ReLU MLP (4 layers, 1024 hidden dim, 1 output dim): Total number of trainbale parameters is 6,302,721
>
> Note that the above model architectures were determined by tuning via grid search on the validation set.
>
> > Since the model’s roots are claimed to be in deep Gaussian processes, I think it would be helpful to see an ablation study comparing deep GPs approximated with random features...
>
> Just to clarify, our model architecture of interchanging between Fourier and linear layers _are determined by deep Gaussian processes_. Perhaps, the only difference from a vanilla random feature expansion of DGPs is how we treated the temporal component, which we pass through a separate network. The reasoning behind this was that normally, spatiotemporal data has different spatial and temporal correlation structures so it made sense to encode them separately before combining their contributions at the end. Perhaps the reviewer is suggesting to conduct an experiment where, we simply take the spatial and temporal coordinates as 3D inputs of a random feature expansion of a DGP? We note that we have conducted comparisons with full DGPs with inference done by doubly stochastic variational inference in our experiments.
>
> > I find the skip connections interesting but somewhat confusing...
>
> Thank you for the suggestion. We were finding that without adding skip connections, sometimes the model lead to poor results, which we suspected was related to the known pathological behaviour of DGPs in Duvenaud et al. 2014. The skip connection we added was simply a remedy as suggested in the same article (theoretical justification is given in Dunlop et al. 2018).
> We illustrate this in the table below, where we compared the losses for our model with and without skip connections (using the dataset in our third experiment):
>
> | DRF Model   | NLPD | CRPS |
> |----------|----------|----------|
> | With skip connections | __0.0063 ± 0.0002__ | __0.0918 ± 0.0021__ |
> | Without skip connections | 0.0078 ± 0.0007 | 0.1092 ± 0.0101 |
>
> We observe that the model without skip connections perform worse on average, with larger model variability. This is due to it sometimes producing erroneous results (see Figure 13 in our updated Appendix).
>
> We're not sure what the reviewer means about the explicit regularisation. Please let us know in more details if you find that this is important.

---

> ### Author Response · Authors · 2024-11-16
> **Continuation of response**
>
> > The paper highlights scalability as a major benefit, but it lacks empirical validation...
>
> Thank you again for the suggestion. We have now conducted an experiment where we trained on $10\\%$ and $1\\%$ subsampled data of our spherical experiment. The results can be found in the table below where we compare our DRF model against SVGP ($3,000$ inducing points, trained for $2$ epochs).
>
> | DRF  | NLPD | CRPS | Time (mins) | SVGP  | NLPD | CRPS | Time (mins) |
> |----------|----------|----------|----------|----------|----------|----------|----------|
> | Full data | __0.0063__ | __0.0927__ | __291.15__ | Full data | 0.0085 | 0.1205 | 345.04 |
> | 10% subsampled data | __0.0079__ | __0.1114__ | __45.61__ | 10% subsampled data| 0.0086 | 0.1200 | 47.46 |
> | 1% subsampled data | 0.1314 | 0.6490 | __17.82__ | 1% subsampled data | __0.0161__ | __0.1435__ | 18.10 |
>
> Our results show that, as expected, the DRF model achieves better performance with increasing data size, as evidenced by the lower NLPD and CRPS values. We also find that for larger data ($10\\%$ and $100\\%$), DRF performs better than SVGP due to its ability to capture finer and finer scale features as we increase data resolution (note that the performance of SVGP does not change much by increasing the data from $10\\%$ to $100\\%$). On the other hand, for smaller data proportions ($1\\%$), the simpler inductive bias in the SVGP helps to generalise better, resulting in better performance compared to DRF. Please see Figure 12 in our updated appendix for qualitative results.
>
> Regarding the ablation with respect to model width, we are currently running experiments and will get back to you as soon as we obtain the results.
>
> > Lastly, given the reliance on random sampling for the random Fourier features, I want to see the variability in the model’s performance. Not on a UQ sense but evaluating the model across multiple runs with varied random seeds.
>
> We actually do report variability of our results across multiple random seeds. This is indicated by $ \mathrm{[mean]} \pm \mathrm{[stddev]}$ across five random seeds in our results in Tables 1 and 2. Please let us know if this is not what you were looking for.

---

> > ### Author Response · Authors · 2024-11-20
> > **Scalability with model width**
> >
> > We have now performed ablations with respect to model width (i.e., number of random features per layer), with results as shown below. In our first experiment. we fixed the bottleneck size $B$ per layer and changed the number of hidden units $H$ between the bottlenecks. The model depth is fixed to $3$. The latter corresponds to the number of per layer random features. We find that generally, more random features (i.e., larger $H$) lead to improved results as we can see in the table below.
> >
> > | DRF Model   | NLPD | CRPS |
> > |----------|----------|----------|
> > | $H=250,  B=128$| 0.0065 ± 0.0001 | 0.0951 ± 0.0006 |
> > | $H=500,  B=128$| 0.0064 ± 0.0001 | 0.0944 ± 0.0009 |
> > | $H=1000, B=128$ | __0.0063 ± 0.0002__ | __0.0918 ± 0.0021__ |
> >
> > We have also conducted an ablation with respect to the bottleneck size $B$ with fixed hidden size $H=1000$:
> >
> > | DRF Model   | NLPD | CRPS |
> > |----------|----------|----------|
> > | $H=1000, B=64$| 0.0067 ± 0.0003 | 0.0969 ± 0.0030 |
> > | $H=1000, B=32$| 0.0066 ± 0.0001 | 0.0969 ± 0.0006 |
> > | $H=1000, B=128$ | __0.0063 ± 0.0002__ | __0.0918 ± 0.0021__ |
> > | $H=1000, B=1000$ | 0.0065 ± 0.0003 | 0.0932 ± 0.0035 |
> >
> > We similarly find that larger bottleneck sizes tend to lead to better results too, but only up to a certain point. When we make the bottlenecks size too large (e.g. taking $B=1000$), we find that the results degrade, with hyperparameter tuning becoming more unstable. Theoretically, it is understood that taking too large a bottleneck size could be detrimental since the process would converge to a Gaussian process in the infinite bottleneck size limit (see Pleiss et al. 2021), losing the flexibility of deep GPs.

---

> > > ### Comment · Reviewer_VtAL · 2024-11-21
> > >
> > > Thank you for your detailed response.
> > >
> > > - To clarify, does your current ReLU MLP include any form of positional encoding or embedding similar to Fourier Features [1] or NeRF [2]? If not, I recommend conducting experiments to compare your model with Tancik et al.’s Fourier Feature Network (FFN) as a baseline.
> > >
> > > - Thank you for providing the ablation results. The observation that increasing the number of hidden units \( H \) generally improves performance and increases variation makes sense and aligns with expectations. However, I noticed that the performance with respect to bottleneck size \( B \) does not appear to follow a clear monotonic trend. This could suggest that the current grid search or experimental setup might not fully capture the optimal configurations. While I am not suggesting additional experiments at this stage, refining the search space or conducting more extensive runs in the future could help ensure more consistent results.
> > >
> > > [1] Tancik, M., Srinivasan, P., Mildenhall, B., Fridovich-Keil, S., Raghavan, N., Singhal, U., Ramamoorthi, R., Barron, J. and Ng, R., 2020. Fourier features let networks learn high frequency functions in low dimensional domains. Advances in neural information processing systems, 33, pp.7537-7547.
> > > [2] Mildenhall, B., Srinivasan, P.P., Tancik, M., Barron, J.T., Ramamoorthi, R. and Ng, R., 2021. Nerf: Representing scenes as neural radiance fields for view synthesis. Communications of the ACM, 65(1), pp.99-106.

---

> > > > ### Author Response · Authors · 2024-11-22
> > > > **Comparison with Fourier features network**
> > > >
> > > > Thank you for clarifying the points.
> > > >
> > > > - For our ReLU MLP baseline, we do not use positional encoding, but we agree with the reviewer that this should be a better baseline to consider. We've now ran an experiment on our first dataset with the Fourier Feature Network (FFN) and get the following results compared to DRF:
> > > >
> > > > | Model  | NLL | CRPS | RMSE |
> > > > |----------|----------|----------|----------|
> > > > | DRF | __13.590 ± 4.899__ | __0.046 ± 0.005__ | __0.135 ± 0.006__ |
> > > > | FFN | 126.869 ± 111.178 | 0.072 ± 0.008 | 0.153 ± 0.005 |
> > > >
> > > > We can still see that DRF still performs better and even qualitatively, we find that DRF is better able to capture the fine scale patterns present in the data. Additionally, we find that the ensembles produced by FFN has very small variance, explaining the poor uncertainty quantification metrics. We believe that the inductive bias of stationary GPs in DRF is helping to produce better calibrated uncertainties.
> > > >
> > > > - Thank you for the suggestion. Yes, we agree that a more extensive search on the bottleneck size can squeeze out better performance.
> > > >
> > > > Should the reviewer have any more suggestions to help improve the rating of this work, we would be happy to try them in the remaining time.

---

> > > > > ### Comment · Reviewer_VtAL · 2024-11-22
> > > > >
> > > > > Thank you for addressing my comments and questions. I plan to increase the score.

---

> > > > > > ### Author Response · Authors · 2024-11-22
> > > > > > **Thank you**
> > > > > >
> > > > > > Thank you very much. We appreciate all your feedbacks that helped to improve the quality of the work. We will add these new results in our updated manuscript.

---

### Author Response · Authors · 2024-11-27
**Updated manuscript**

We have now uploaded a revised version of our manuscript, which should address most of the points raised by the reviewers with additional ablation studies in the appendices (attached as supplementary material). If the reviewers have further comments and suggestions on our revisions, we are happy to engage in discussions until the end of the discussion period.

---

> ### Comment · Reviewer_jnCy · 2024-12-02
>
> Thank you for revising the manuscript and providing detailed responses to the comments. I have updated my confidence score from 3 to 4 while maintaining my overall rating of 8.

---

### Meta-Review · Area_Chair_3zXA · 2024-12-19

**Metareview:**

The paper 'Deep Random Features for Scalable Interpolation of Spatiotemporal Data' was reviewed by 3 reviewers who gave it an average score of 6.33 (3+8+8). The reviewers appreciated the versatility of the considered methods, the coverage of background, and the presentation. One of the reviewers raised concerns about the modelling assumptions, coverage of recent related works, and called for more discussion. The authors addressed the majority of the concerns during the rebuttal phase. Overall, the paper appears to lean towards acceptance, and it should be of interest to the ICLR community.

**Additional Comments On Reviewer Discussion:**

The authors posted rebuttals that addressed the reviewer concerns. Two out of three reviewers were active during the discussion, with the only negative reviewer not being active. The average score increased from 5.67 -> 6.33 during the discussion.

---

### Decision · Program_Chairs · 2025-01-22

Accept (Poster)